# First Things First: Teaching MLLM Agents to Prioritize Must-Haves before Nice-to-Haves

## Abstract

Recent progress in multimodal large language models (MLLMs) has fueled significant enthusiasm in their potential to act as autonomous agents for real-world tasks. However, scenarios requiring agents to fulfill users' complex, structured requirements remain largely underexplored. In this work, we examine reasoning tasks under three distinct requirement scenarios, each defined by the feasible solution set delineated by must-have and nice-to-have requirements: (i) Must-have requirements uniquely determine a unique feasible solution; (ii) Multiple candidate solutions satisfy the must-have requirements and are prioritized via the nice-to-have requirements; and (iii) No candidate solution satisfies the must-have requirements, in which case the agent should abstain from generating a response. We evaluate state-of-the-art MLLMs on 3,649 carefully constructed problems that reflect realistic service scenarios, including e-commerce platforms, booking systems, and map-based or ride-hailing applications. Our evaluation reveals that existing MLLMs exhibit catastrophic failures in all scenarios. Specifically, these models frequently misinterpret task requirements, violate must-have requirements, and produce invalid solutions. To address this critical gap, we propose First Things First Reinforcement Learning (FTF-RL) that explicitly optimizes reasoning over multi-priority user requirements. Experimental results show that our method substantially improves the task success rate compared to strong baselines. Moreover, FTF-RL yields general effectiveness on popular logical and mathematical reasoning tasks, including LogicVista, MathVision, and MathVista. Our finding suggests that enhancing requirement comprehension provides a simple yet effective pathway toward improving the broad generalization of MLLMs. Code and evaluation data are available at *anonymity*.

## 1 Introduction

Recent advances in multimodal agents have shown promising results in automating complex tasks such as booking flights, reserving hotels, or navigating graphical user interfaces (GUI) through natural language instructions (Xie et al., 2024a;b; Cheng et al., 2025; Guo et al., 2025). These systems are typically evaluated in settings where users provide unambiguous instructions, the agent simply needs to execute the command accurately (Jiang et al., 2024b; Rawles et al., 2025; Zhang et al., 2025c). While impressive, such scenarios represent an idealized setting: the intent is clear, the requirements are minimal, and there is little ambiguity about what constitutes a correct solution.

In real-world service scenarios, however, user requirements are rarely so simple (Wen et al., 2024; Ji et al., 2025; Liu et al., 2025). Requests are often multifaceted, containing both **must-have** requirements and **nice-to-have** requirements. Current agents often struggle in these settings because they lack a mechanism to prioritize hard requirements over soft requirements, and thus frequently overfit to all expressed conditions, returning either infeasible solutions or incorrect results. For example, when asked to "*book a non-smoking hotel for two people, preferably with breakfast included*", a typical agent might return a hotel that includes breakfast but fails to ensure the non-smoking requirement, violating the user's core intent. Conversely, when given conflicting requirements, such as "*book the cheapest hotel with two bedrooms, but also with a sea view if possible*", agents often fail to resolve the trade-off and either refuse to answer or return an irrelevant option.

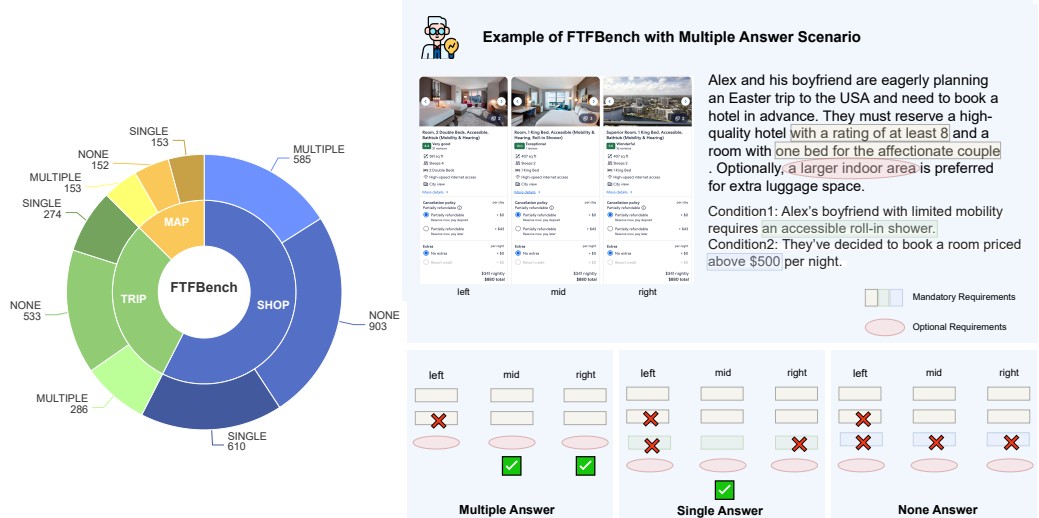

Figure 1: Overview of our proposed FTF-BENCH, which designed to evaluate requirement-aware reasoning in MLLMs. FTF-BENCH spans realistic service domains including e-commerce, booking platforms, and map or ride-hailing interfaces, with user queries expressed through both must-have and nice-to-have requirements. Tasks are categorized into three settings: Single-Answer (a unique valid candidate satisfies all must-haves), Multiple-Answer (several candidates meet the must-haves and must be ranked via nice-to-haves), and Unanswerable (no candidate satisfies all must-haves, requiring abstention).

To systematically study this problem, we construct the First Things First Benchmark (FTF-BENCH) to evaluate the capabilities of MLLMs in parsing and reasoning under a clear necessity hierarchy. FTF-BENCH contains 3,649 image–requirement pairs from realistic e-commerce, booking, and maps or ride-hailing interfaces. For each instance, we synthesize a colloquial user request with must-have and nice-to-have components, verified by human annotators for correctness and clarity. All tasks are grouped into three settings that mirror real-world scenarios: (i) must-have requirements uniquely determine a single valid match; (ii) multiple candidates satisfy all must-have requirements and the system must rank them using soft requirements; and (iii) no candidate satisfies the hard requirements (Figure 1). This design supports automatic evaluation and isolates where MLLMs fail to identify requirements versus where they fail to trade off requirements (Section 3).

We further introduce a reinforcement learning approach designed to enhance requirement-aware reasoning. Standard SFT (Wang et al., 2023; Peng et al., 2023) and RLHF (Ouyang et al., 2022; Yuan et al., 2023) improve general compliance yet offer little direct supervision for identifying necessity and ordering requirement satisfaction. We propose First Things First Reinforcement Learning (FTF-RL) that trains MLLMs with a multi-objective reward that checks well-formed outputs, final answer correctness, and accurate classification of requirements into must-have and nice-to-have, while encouraging transparent intermediate reasoning. The reward structure guides the MLLMs to first ground user intent in an explicit requirement parse and then reason about candidates, rather than mixing these steps in a brittle manner (Section 4).

We systematically evaluate current MLLMs on FTF-BENCH and find that catastrophic failures persist widely across domains. These MLLMs often over-constrain by elevating requirements to complex rules or under-constrain by ignoring mandatory conditions, which leads to incorrect selections. Errors are most severe in the multiple-answer and unanswerable settings where MLLMs must both respect necessity and resolve trade-offs or abstain. When we supply gold requirement labels as an upper-bound setting, accuracy improves substantially, and the gains are especially pronounced for open-source models, which shows that better requirement comprehension alone can substantially unlock the potential of MLLMs (Section 5.2).

Building on the benchmark diagnosis, we further adopt the proposed FTF-RL to train MLLMs for stronger requirement-aware reasoning. We apply FTF-RL to Qwen 2.5 VL (Bai et al., 2025) of

different sizes and observe consistent improvements on FTF-BENCH across single-answer, multiple-answer, and unanswerable settings. Beyond FTF-BENCH, models fine-tuned with FTF-RL show gains on other reasoning benchmarks, indicating that learning to parse and prioritize requirements transfers to broader reasoning skills. An ablation that removes the requirement-classification term reduces accuracy, especially in multiple-answer and unanswerable cases, confirming the value of explicit supervision over necessity. We call for greater attention to the role of requirement-aware reasoning (Section 5.3.1).

## 2 RELATED WORK

### 2.1 MULTIMODAL REASONING

Research on reasoning in MLLMs has advanced through Multimodal Chain-of-Thought (MCoT) and RL approaches. To inject more procedural structure, early CoT-based approaches (Li et al., 2025) decomposed reasoning into stages such as perceptual summarization, localized grounding, and fine-grained verification. However, their rigid pipeline structure limited adaptability to diverse tasks. Subsequent variants adopted more flexible decomposition of reasoning: for example, Cantor (Gao et al., 2024) explicitly partitions the model's processing into perception and decision steps, while TextCoT (Luan et al., 2024) refines reasoning by zooming from global descriptions to local crop analyses. For specialized reasoning, Fei et al. (2025) proposed Video-of-Thought (VoT), which breaks down video clips into temporally ordered segments to improve action prediction within video-chat benchmarks.

Reinforcement learning has emerged to elicit deeper reflection and optimize reasoning quality. RL frames reasoning as a Markov Decision Process (MDP), optimizing trajectories via rewards. DeepSeek-AI et al. (2025) used verifiable rewards (e.g., math correctness), which Pan et al. (2025) extended to MLLMs with MedVLM-R1. For spatial reasoning, Ouyang (2025) used spatial consistency rewards to boost 6D reasoning accuracy. Recent RL advances like StepGRPO added intermediate rewards, enhancing logical consistency on R1-VL (Zhang et al., 2025a). However, most prior RL approaches focus on pushing the upper bound of VLM reasoning performance, without explicitly ensuring that the model fully understands and decomposes the requirements of the question. To address this gap, we design a requirement-aware reward, resulting in reasoning chains that are both accurate and aligned with the task intent.

### 2.2 INSTRUCTION FOLLOWING BENCHMARKS IN MLLMS

A variety of benchmarks have been introduced to evaluate multimodal large language models (MLLMs) across multiple facets. These encompass assessments of general knowledge (Liu et al., 2024; Yue et al., 2024), document understanding (Yue et al., 2024; Fu et al., 2023), perceptual reasoning (Hu et al., 2020), multi-image comprehension (Yu et al., 2024), and instruction following (IF) (Qian et al., 2025; Bitton et al., 2023).

Several benchmarks have been proposed to quantify the instruction-following capability of large language models (Jiang et al., 2024a; Zhang et al., 2025b; Hida et al., 2024). For instance, LIF-Bench (Wu et al., 2025) limits its scope to instructions whose fulfillment can be verified automatically, which enhances reproducibility and model differentiation. InfoBench (Qin et al., 2024) decomposes prompts into atomic sub-instructions and computes the Decomposed Requirements Following Ratio (DRFR), thereby granting partial credit for each satisfied micro-requirement. Benchmarks like MIA-Bench (Qian et al., 2025) and VisIT-Bench (Bitton et al., 2023) adopt GPT-4 for both question generation and evaluation in instruction following settings. However, existing instruction-following benchmarks implicitly treat all directives as equally important. In contrast, our proposed FTF-BENCH introduces a necessity hierarchy to instruction-following tasks, evaluating whether models prioritize mandatory requirements before optional ones rather than treating all directives as equally critical.

Table 1: Distribution of FTF-BENCH across different task scenarios.

| Image Scene | Single-Answer | Multiple-Answer | Unanswerable | Total |
|---|---|---|---|---|
| E-commerce Platforms | 585 (27.9%) | 903 (43.0%) | 610 (29.1%) | 2098 |
| Booking Services | 286 (26.2%) | 533 (48.8%) | 274 (25.1%) | 1093 |
| Maps & Ride-Hailing | 153 (33.4%) | 152 (33.2%) | 153 (33.4%) | 458 |

## 3 INTRODUCING FIRST THINGS FIRST BENCHMARK

### 3.1 OVERVIEW

Real-world intelligent agents, such as customer-service chatbots and virtual assistants, must process user requests involving multiple requirements, where satisfying must-to-haves before nice-to-haves is critical for success. Existing multimodal benchmarks mainly assess generic instruction following and rarely measure whether models can identify and prioritize must-have requirements before optimizing for requirements. To address this gap, we introduce FTF-BENCH, a comprehensive evaluation suite for assessing MLLMs' ability to understand and reason over complex, multi-priority user requirements.

FTF-BENCH covers three representative real-world domains: e-commerce platforms, booking services (e.g., hotel, flight, train), and maps & ride-hailing applications (Figure 1). These images span diverse interface layouts and interaction contexts. Tasks in FTF-BENCH are categorized based on the relationship between candidates and mandatory requirements. Single-answer tasks have a unique option that satisfies all mandatory conditions. Multi-answer tasks have multiple valid options, requiring models to consider optional requirements and trade-offs to select the best choice. Unanswerable tasks contain no valid options, testing whether models can detect infeasible requests and respond appropriately. Table 1 details the distribution of samples across these combinations. In total, the dataset comprises 3,649 samples, of which e-commerce platforms contribute 2,098 samples (57.5%), booking services 1,093 samples (30.0%), and maps & ride-hailing 458 samples (12.5%).

### 3.2 TASK FORMULATION

In this section, we formalize the definition of our benchmark tasks, which are designed to evaluate three representative scenarios *unique-answer*, *multiple-answer*, and *unanswerable*. Each sample is represented as:

$$\mathcal{D} = \{x_i\}_{i=1}^N, \qquad x = (I, \mathcal{O}, \mathcal{R}^+, \mathcal{R}^-), \tag{1}$$

where $I$ is the input image, $\mathcal{O} = \{o_1, \ldots, o_M\}$ is the set of $M$ candidate objects appearing in $I$, $\mathcal{R}^+$ is the set of must-have requirements, and $\mathcal{R}^-$ is the set of nice-to-have requirements. Each requirement $r \in \mathcal{R}^+ \cup \mathcal{R}^-$ is a logical predicate evaluated on a candidate $o \in \mathcal{O}$. The satisfaction of a candidate $o$ with respect to a requirement $r$ is given by:

$$\mathrm{sat}(o, r) \in \{0, 1\}, \qquad \mathrm{sat}(o, r) = 1 \Leftrightarrow o \text{ satisfies } r. \tag{2}$$

A candidate satisfies all must-have requirements if

$$\mathrm{sat}^+(o) := \prod_{r \in \mathcal{R}^+} \mathrm{sat}(o, r) = 1, \tag{3}$$

and the subset of candidates that meet all must-have requirements is:

$$\mathcal{S}^+ = \{o \in \mathcal{O} \mid \mathrm{sat}^+(o) = 1\}. \tag{4}$$

The first scenario, single-answer, is defined by $|\mathcal{S}^+| = 1$, meaning exactly one candidate satisfies all must-have requirements.

The second scenario, multiple-answer, corresponds to $|\mathcal{S}^+| \geq 2$. In this case, candidates must be further distinguished using the nice-to-have requirements $\mathcal{R}^-$. Each optional requirement $r_j^-$ has an associated priority $p(r_j^-) \in \mathbb{N}^+$. Let $\pi(\mathcal{R}^-) = (r_{(1)}^-, r_{(2)}^-, \ldots, r_{(L)}^-)$ denote the priority-ordered sequence. Starting from $\mathcal{S}_0 = \mathcal{S}^+$, candidates are iteratively filtered according to $\mathcal{S}_k = \{o \in \mathcal{S}_{k-1} \mid$

$\text{sat}(o, r^-_{(k)}) = 1\}$. After applying all optional requirements, the remaining candidate set $\mathcal{S}_L$ is used to determine the final answer.

The third scenario, unanswerable, corresponds to $\mathcal{S}^+ = \varnothing$, meaning no candidate satisfies all must-have requirements. Any prediction of a candidate option in this case is counted as an error.

### 3.3 BENCHMARK CONSTRUCTION

To construct a high-quality dataset for training and evaluation, we develop a large-model-driven pipeline that systematically generates image-requirement pairs. The pipeline comprises the following core stages: Image Collection, Requirement Generation, Colloquial Requirement Expression, and Human Verification. Several representative examples are presented in Appendix A.

**Image Collection.** We first assemble a diverse image corpus to capture realistic service-oriented scenarios. For each domain, volunteers collect screenshots from multiple applications, sampling different interface states (e.g. homepages, product pages, shopping carts, search results). This design ensures broad coverage of everyday usage contexts, and yields a dataset reflecting high diversity and structural complexity.

**Requirement Generation.** For each image, we leverage LLMs to generate both requirements and ground-truth answers. We design prompt templates to align precisely with three task types (single-answer, multi-answer, unanswerable), ensuring that each requirement is grounded in the image content. Specifically, we produce a labeled requirement set that separates must-have ($\mathcal{R}^+$) from nice-to-have ($\mathcal{R}^-$) requirements, and for multi-answer cases we assign an explicit priority order over $\mathcal{R}^-$ to support ranking among candidates.

**Colloquial Requirement Expression.** We then convert structured requirements into natural, conversational queries, better emulating real user behavior. In practice, users rarely frame needs in formal subclauses; rather, they pose a single coherent question that interleaves must-have and nice-to-have requirements. To simulate this, we prompt the model to rewrite requirements as context-aware, fluent utterances while preserving the priority hierarchy. This step preserves the explicit must-have/nice-to-have distinction in a single utterance so that downstream evaluation can still reflect the necessity-first objective rather than generic instruction following.

**Human Verification.** Finally, we apply human verification to secure benchmark integrity. Four trained annotators independently inspect each image–query pair, validating requirement correctness, answer consistency, and the separation between mandatory and optional constraints. Any disagreement is resolved via discussion, producing a high-quality, standardized dataset that supports robust evaluation. During checking, annotators ensure that each instance conforms to one of the three settings so that the final benchmark faithfully measures requirement-aware reasoning. We provide several prominent error types in Appendix D.

## 4 FIRST THINGS FIRST REINFORCEMENT LEARNING

### 4.1 MULTI-OBJECTIVE REWARD FUNCTIONS

Reward models play a crucial role in RL, as they directly determine the optimization signal that guides policy improvement. Recent advances, such as DeepSeek-R1 (DeepSeek-AI, 2025), have demonstrated that verifiable reward functions can substantially enhance the reasoning ability of MLLMs. Inspired by this success, we design a rule-based, multi-objective reward function to evaluate both requirement understanding and visual reasoning capabilities. This design ensures that the model not only produces correct final answers but also generates interpretable intermediate steps, thereby improving generalization. Specifically, our reward framework assesses model outputs along four key dimensions: format compliance, final-answer correctness and requirement classification accuracy, which distinguishes between must-have and nice-to-have requirements (Figure 2).

**Format Reward.** The format reward $R_{\text{format}}$ verifies whether the model output strictly follows the predefined XML-style schema. Specifically, we check three components: (i) whether the requirement classification is enclosed within `<requirements>...</requirements>` tags and

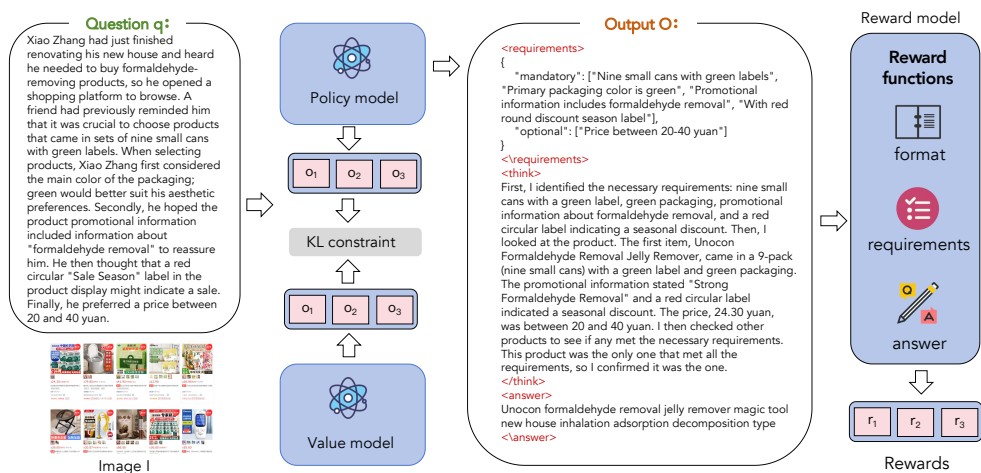

Figure 2: **FTF-RL framework.** Given an input image $I$ and question $q$, we first sample $G$ candidate outputs $\{o_i\}_{i=1}^{G}$ from the old policy model. Then we compute a reward $r_i$ for each $o_i$ using our proposed multi-objective reward function (see Section 4.1 for details). Finally, we optimize the current policy model by maximizing $A_i$, while regularizing the update using KL divergence between $\pi_\theta$ and the reference policy model $\pi_{\text{ref}}$ to keep the updated policy close to the reference policy.

expressed as a valid JSON object separating `must_have` and `nice_to_have` fields; (ii) whether the intermediate reasoning process is included in `<think>...</think>` tags; and (iii) whether the final answer is provided in `<answer>...</answer>` tags. The reward is computed as a step-wise function:

$$R_{\text{format}}(y) = \begin{cases} 1, & \text{if all three components are valid and well-formed,} \\ 0, & \text{if none of the components follow the required format.} \end{cases} \quad (5)$$

This reward encourages the model to produce fully structured, machine-parsable, and human-readable outputs, facilitating both automatic evaluation and transparent error analysis.

**Accuracy Reward.** The accuracy reward $R_{\text{answer}}$ measures whether the model's final prediction matches the ground-truth answer. Since our tasks are formulated as fill-in-the-blank questions, we employ a powerful MLLM as a judging model to robustly compare the model-generated answer against the reference solution. This approach mitigates surface-form mismatch issues (e.g., synonyms, equivalent expressions). Formally, for a given question $q$, model output $y$, and ground-truth answer $a^*$, we define:

$$R_{\text{answer}}(y, a^*) = \begin{cases} 1, & \text{if Judger}(q, \text{Ans}(y), a^*) = \texttt{True}, \\ 0, & \text{otherwise,} \end{cases} \quad (6)$$

where $\text{Ans}(y)$ extracts the content between the `<answer>...</answer>` tags from the model output, and $\text{Judger}(\cdot)$ is the MLLM-based evaluation function that returns `True` if the predicted answer is semantically equivalent to the ground-truth answer $a^*$.

**Requirement Reward.** The requirement reward $R_{\text{requirement}}$ evaluates the model's ability to correctly identify and classify user requirements into `must_have` and `nice_to_have` categories. This component is motivated by the observation that many reasoning failures arise not from incorrect computation, but from misinterpretation of user intent: models often over-constrain by treating nice-to-have requirements as hard requirements, or under-constrain by ignoring must-have requirements, leading to invalid or suboptimal answers. By explicitly rewarding correct requirement classification, we encourage the model to faithfully represent user intent before performing reasoning, thereby reducing downstream reasoning errors and improving decision quality.

Formally, let $J_{\text{pred}}$ denote the JSON object produced by the model within the `<requirements>` tags, and $J_{\text{gold}}$ the ground-truth classification. We define a normalized F1-based reward:

$$R_{\text{requirement}}(y, J_{\text{gold}}) = \frac{1}{2}\big(\text{F1}(\mathcal{R}_{\text{pred}}^+, \mathcal{R}_{\text{gold}}^+) + \text{F1}(\mathcal{R}_{\text{pred}}^-, \mathcal{R}_{\text{gold}}^-)\big), \tag{7}$$

where $\text{F1}(\cdot, \cdot)$ computes the F1-score between the predicted and gold requirement sets for each category. This design provides a soft, differentiable reward signal that penalizes both false positives (e.g., over-constraining) and false negatives (e.g., ignoring essential requirements).

## 5 EXPERIMENTS

### 5.1 SETUP

We evaluate seven widely used MLLMs that represent the current frontier on FTF-BENCH. The proprietary group includes Gemini 2.5 Pro (Gemini, 2025), GPT-5 (OpenAI, 2025a), GPT-o3 (OpenAI, 2025b), and Doubao 1.6 Seed (ByteDance Seed, 2025). The open-source group covers Qwen2.5-VL (Bai et al., 2025), LLaMA-4 (Meta AI, 2025) and Claude-Sonnet-4.5(Anthropic, 2025). For Qwen2.5-VL, we report results for the 7B, 32B, and 72B checkpoints to examine scaling effects. We also report results on four variants of LLaVA (Liu et al., 2023). To study the impact of our method, we apply FTF-RL to Qwen2.5-VL-3B, LLaVA-OneVision-7B and LLaVA-1.5-13B and evaluate the fine-tuned MLLMs alongside their instruction-tuned baselines on FTF-BENCH and additional reasoning suites.

For benchmark evaluation, we conduct experiments on all 3,649 instances of FTF-BENCH as illustrated in Figure 1. Detailed prompts used in both Direct and Upper settings are provided in Appendix B. For reinforcement learning, we train MLLMs using 90% of the benchmark data and reserve the remaining 10% for evaluation. We have verified that this 10% evaluation subset is representative of the full benchmark in Appendix K. All reinforcement learning experiments are conducted under a unified evaluation to achieve fair comparison. We adopt GRPO (Shao et al., 2024) with a KL-penalty coefficient of $10^{-2}$ to stabilize policy updates. Training is conducted with a global batch size of $128$ and a learning rate of $10^{-6}$, using AdamW optimization and full-shard FSDP for efficient distributed training. During rollout, we use a temperature of $1.0$ and sample $n = 5$ candidate responses to encourage exploration.

### 5.2 EVALUATION RESULTS ON FTF-BENCH

We conduct a comprehensive evaluation of current MLLMs on FTF-BENCH under two input settings. In the Direct setting, MLLMs read the original colloquial user request and must infer both the requirement hierarchy and the final decision. In the Upper setting, we paraphrase the same request into gold requirement labels that separate must-haves from nice-to-haves and feed these labels to MLLMs as structured guidance, which serves as an empirical upper bound for performance with perfect requirement understanding. We provide the main results in Table 2.

Across most scenarios, Upper exceeds Direct, confirming that the primary source of error is not visual perception alone but the failure to correctly parse and prioritize requirements from natural language. Nearly all MLLMs show a noticeable decline when directly interpreting the original user text. The gap is particularly severe in the multiple-answer and unanswerable scenarios. For multiple-answer tasks, candidate options often contain several plausible distractors, and only by reasoning with nice-to-have conditions can the optimal choice be determined. Without explicit requirement decomposition, MLLMs struggle to resolve these fine-grained trade-offs. For unanswerable tasks, where no candidate satisfies the must-have requirements, MLLMs should reject all options. However, most MLLMs still attempt to produce an answer, revealing a tendency to over-accommodate user prompts even when abstention is the correct strategy. These observations expose a fundamental weakness in requirement-aware reasoning.

Another observation is the large gap between model performance and the upper bound. While proprietary MLLMs maintain relatively high accuracy under both settings, the difference between their direct and upper results remains non-trivial. For open-source MLLMs, the discrepancy is even

Table 2: Evaluation results of current MLLMs on FTF-BENCH across three task types. Upper denotes the performance when MLLMs are given golden requirements, while Direct reports accuracy when MLLMs directly answer from the original prompt without additional guidance.

| Models | Single-Answer | | Multiple-Answer | | Unanswerable | | Average | |
|---|---|---|---|---|---|---|---|---|
| | Upper | Direct | Upper | Direct | Upper | Direct | Upper | Direct |
| *Proprietary MLLMs* | | | | | | | | |
| Gemini-2.5-pro[†] | **88.89** | 86.91 | **84.20** | 82.26 | **81.72** | 78.55 | **84.26** | 81.75 |
| GPT-5[†] | 83.57 | 85.21 | 80.73 | 78.10 | **82.95** | 80.24 | 82.49 | 80.86 |
| GPT-o3[†] | **82.67** | 77.78 | 80.41 | 80.03 | 83.31 | 83.09 | 82.33 | 79.68 |
| Doubao-1.6-seed[†] | 80.09 | 78.66 | 81.70 | 76.99 | 83.80 | 84.05 | 82.24 | 80.59 |
| Claude-Sonnet-4.5[†] | **77.52** | 75.37 | **73.89** | 71.94 | 80.98 | 82.49 | **78.01** | 77.51 |
| *Open-Source MLLMs* | | | | | | | | |
| LLaMA-4 | **58.36** | 55.48 | **55.93** | 54.45 | **44.43** | 36.94 | **55.03** | 51.99 |
| Qwen2.5-VL-7B-Instruct | **61.16** | 22.99 | **58.25** | 20.10 | **47.59** | 18.64 | **57.69** | 21.05 |
| Qwen2.5-VL-32B-Instruct | **70.66** | 69.25 | **70.55** | 70.17 | **53.08** | 50.58 | **67.72** | 66.57 |
| Qwen2.5-VL-72B-Instruct | **71.26** | 35.76 | **74.75** | 35.12 | **83.09** | 29.62 | **77.60** | 34.48 |
| LLaVA-OneVision-7B | **45.28** | 19.15 | **43.17** | 18.22 | **42.36** | 21.78 | **43.38** | 19.73 |
| LLaVA-NEXT-34B | **50.31** | 27.27 | **50.20** | 20.21 | **51.40** | 25.83 | **50.76** | 24.11 |
| LLaVA-NeXT-13B | **48.65** | 26.92 | **46.89** | 18.34 | **46.72** | 33.95 | **47.25** | 26.13 |
| LLaVA-1.5-13B | **48.70** | 26.98 | **46.94** | 16.39 | **42.77** | 34.00 | **45.72** | 25.28 |

more striking. In particular, Qwen2.5-VL and LLaMA-4 exhibit substantial improvements once gold requirements are provided, suggesting that their reasoning pipeline is hampered less by visual limitations than by misinterpretation of complex user intent.

Interestingly, scaling effects are not monotonic. Within Qwen2.5-VL, the 7B and 72B models fail catastrophically in direct reasoning, whereas the 32B variant demonstrates comparatively strong capability. We interpret this as evidence of two distinct failure modes. The smaller 7B model appears to lack the basic capacity to reliably distinguish must-haves from nice-to-haves, which leads to frequent violations of core requirements. The largest 72B model, while more powerful in general, shows an opposite failure pattern. Its tendency to overfit to the surface form of user prompts results in excessive alignment with every expressed condition, elevating optional requirements to mandatory status and thereby producing infeasible outputs. In contrast, the 32B model achieves a more balanced handling of requirement prioritization, revealing that scale alone does not guarantee progress and that training dynamics strongly influence requirement-aware reasoning. To further characterize how MLLMs distinguish between must-have and nice-to-have requirements in the Direct setting, we also report requirement-classification confusion matrices for representative MLLMs in Appendix G.

### 5.3 IMPROVE REQUIREMENT-AWARE REASONING BY FTF-RL

#### 5.3.1 MAIN RESULTS

We further improve requirement-aware reasoning by applying FTF-RL to open-source Qwen2.5-VL. We randomly sample 90% of FTF-BENCH for training and hold out the remaining 10% for evaluation. Comparison results are presented in Table 3.

**Both the 7B and 3B variants deliver strong gains, demonstrating that FTF-RL is highly effective at strengthening the requirement-aware reasoning capabilities of MLLMs.** The most remarkable results appear in the multiple-answer scenario. Qwen2.5-VL-7B-Instruct improves by more than 26% after reinforcement learning, showing that the method enables MLLMs to reliably separate must-have conditions from nice-to-have preferences and make the correct choice even when several confusing candidates are present. The single-answer and unanswerable settings also benefit, with consistent improvements observed across scales.

Table 3: Impact of reinforcement learning on different MLLMs

| Model | Single-Answer | Multiple-Answer | Unanswerable | Average |
|---|---|---|---|---|
| Qwen2.5-VL-3B-Instruct | 48.55 | 33.13 | 32.79 | 38.95 |
| + FTF-RL | **56.52**↑7.97 | **41.10**↑7.97 | **34.42**↑1.63 | **45.85**↑6.90 |
| Qwen2.5-VL-7B-Instruct | 46.38 | 32.52 | **44.26** | 39.78 |
| + FTF-RL | **57.97**↑11.59 | **58.90**↑26.38 | 42.62↓1.64 | **55.80**↑16.02 |
| LLaVA-OneVision-7B | 45.82 | 33.69 | 42.87 | 44.09 |
| + FTF-RL | **52.51**↑6.69 | **40.15**↑6.46 | **48.79**↑5.92 | **50.48**↑6.39 |
| LLaVA-1.5-13B | 49.23 | 37.51 | 43.19 | 46.64 |
| + FTF-RL | **57.81**↑8.58 | **43.74**↑6.23 | **49.05**↑5.86 | **52.87**↑7.23 |

Table 4: Effect of FTF-RL on model generalization across various reasoning benchmarks.

| Model | LogicVista | MathVision | MathVista | InfoQA |
|---|---|---|---|---|
| Qwen2.5-VL-7B-Instruct | 43.40 | 24.67 | 60.80 | 65.35 |
| + FTF-RL | **47.43**↑4.03 | **25.65**↑0.98 | **61.30**↑0.05 | **67.81**↑2.46 |
| Qwen2.5-VL-3B-Instruct | 36.91 | 23.03 | **40.70** | 37.62 |
| + FTF-RL | **40.49**↑3.58 | **24.34**↑1.31 | 39.50↓1.20 | **39.93**↑2.31 |
| LLaVA-1.5-13B | 29.23 | 11.12 | 27.64 | 41.57 |
| + FTF-RL | **35.61**↑6.38 | **13.74**↑2.62 | **29.05**↑1.41 | **41.92**↑0.35 |

### 5.3.2 GENERALIZATION ACROSS REASONING BENCHMARKS

To further investigate how requirement-aware reasoning contributes to the general capability of MLLMs, we compare MLLMs trained only on FTF-BENCH with their baselines on other logic and math reasoning benchmarks, including LogicVista (Xiao et al., 2024), MathVision (Wang et al., 2024), and MathVista (Lu et al., 2024). Table 4 presents the results of Qwen2.5-VL models before and after FTF-RL training. We are surprised to find that even without any explicit training on these reasoning benchmarks, the MLLMs exhibit consistent improvements after reinforcement learning on most reasoning benchmarks. This suggests that **requirement-aware reasoning not only strengthens the understanding of complex user intent but also stimulates the general reasoning ability of MLLMs, leading to clear gains across diverse and challenging tasks**. We present two case studies on LogicVista and MathVision in Appendix I.

## 6 CONCLUSION

In this paper, we investigate requirement-aware reasoning in realistic service settings where LLM-based agents are required to prioritize must-haves before nice-to-haves. We first introduce FTF-BENCH to evaluate the requirement-aware reasoning capability across various scenarios with 3,649 constructed instances. Experiments show that current MLLMs frequently misread requirements, violate hard requirements, and output invalid solutions, leading to catastrophic failures in prioritizing requirements. To improve the requirement-aware reasoning capability, we further present a multi-objective RL framework FTF-RL that rewards correct requirement identification, proper ordering of requirement satisfaction, and answer validity. After training, MLLMs perform substantially better and narrow the gap toward the upper bound obtained with requirement labels. We also observe consistent gains on other tasks that require complicated reasoning, which suggests that strengthening requirement comprehension yields broader generalization. We call for more attention to the pivotal role of requirement-aware reasoning in advancing the reasoning reliability of MLLMs.

ETHICAL CONSIDERATIONS

All authors of this work have read and agree to abide by the ICLR Code of Ethics. Our study focuses on evaluating and enhancing requirement-aware reasoning in MLLMs. We call for the widespread deficiencies of current models in prioritizing must-have over nice-to-have requirements and introduce reinforcement learning methods that substantially improve this capability. All experiments are conducted on data we collected through screenshots of publicly available applications, with requirements automatically generated and subsequently verified by trained human annotators. Annotators were compensated fairly at a rate of 1 RMB per data sample, and no personally identifiable or sensitive information is included in the dataset. We confirm that the benchmark does not introduce bias toward any social group, nor does it involve privacy, security, or harmful use concerns. For paper preparation, we only used LLMs for light editing after completing the writing, limited to correcting typos and grammar errors, and did not rely on LLMs for substantive content generation.

REPRODUCIBILITY STATEMENT

We fully commit to the reproducibility of all results reported in this paper. Section 5.1 introduces our experimental setup and evaluation protocols, while Section 3.3 provides the complete benchmark construction pipeline. Appendix B and C include the prompts used in both evaluation settings. We promise to release the complete benchmark and codebase for community use.

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

Table 5: Comparison of two error examples, presented in a column-wise layout for side-by-side inspection.

| Item | Example 1 | Example 2 |
|---|---|---|
| Image |  |  |
| Error Type | Must-have Conflict, No Refusal | Misclassification of Nice-to-have as Must-have |
| User Requirements | **Must-have:** 1 King Bed required. **Must-have:** Must accommodate 4 guests. **Nice-to-have:** Prefer balcony and partial ocean view. | **Must-have:** Price per night $\leq$ \$300. **Must-have:** Must accommodate 2 guests. **Nice-to-have:** Prefer rating $\geq 8.5$. |
| Model Output (Incorrect) | Recommended "Room, 2 Double Beds, Balcony, Partial Ocean View", 377 sq ft, accommodates 4 guests, includes balcony and partial ocean view — claimed to fully meet the request. | "Sorry, no available rooms meet the requirements." |
| Expected Output | The system should reject or indicate no available room meets all must-have requirements. Example: "No room satisfies both the '1 King Bed' and '4 guests' requirements." | The system should treat rating criterion as optional and list all rooms satisfying the price and capacity requirements, possibly ranked by rating. Example: "3 rooms found under \$300 for 2 guests, sorted by rating." |

# A    REPRESENTATIVE EXAMPLES FROM FTF-BENCH

Table 5 presents representative instances from FTF-BENCH, illustrating typical error types made by MLLMs. It shows how models often misclassify must-have versus nice-to-have requirements or fail to properly reject infeasible requests.

# B    EVALUATION PROMPTS

The detailed prompt for evaluating FTF-BENCH in Section 5.2 is shown below:

**Prompts for Evaluating FTF-BENCH**

Given the following task requirements, strictly select the product from the image that best satisfies the criteria. If no product satisfies all the "must-have requirements," explicitly return a refusal. Selection must be based solely on the information provided in the task and visible in the image; no additional reasoning or assumptions are allowed. The rules are as follows:

1. Requirement types:
- "Must-have requirements"
- "Nice-to-have requirements"

2. Must-have requirements:
- A product must satisfy all must-have requirements simultaneously; missing any requirement disqualifies it.
- Any product failing to meet even one must-have requirement is considered ineligible and cannot proceed to further comparison.

3. Priority-based selection of nice-to-have requirements (applied only if more than 2 candidate products remain):
- Only when multiple products satisfy all must-have requirements should nice-to-have requirements be used for further selection.
- Compare products sequentially according to the priority of nice-to-have requirements, from highest to lowest (priority 1 being the highest).
- Select the product that satisfies more of the higher-priority requirements. If there is a tie at one priority level, continue comparing the next lower priority until a unique best product is identified.
- Example: If priority is ordered as 1>2>3..., and Product A satisfies priorities 1, 2, 3 while Product B satisfies priorities 1, 2, 4, 5, Product A should be chosen because priority 3 is higher than 4 or 5.

Output rules:
1. If at least one product satisfies all must-have requirements:
- If only one product qualifies, directly output its name.
- If multiple products qualify, apply the nice-to-have priority rules to select the unique best product and output its name.
2. If no product satisfies all must-have requirements, output the refusal text: "No product satisfies all must-have requirements."

Output format:
- Must output strictly in JSON format and only output the JSON.
- Format:
{"Answer": ""}
- "Answer": Fill in the name of the selected product. If no product qualifies, set the value to "No product satisfies all must-have requirements" (including quotes).

The question is as follows:
<question>
Strictly follow the above rules and base the selection solely on the information provided in the task and visible in the image.

## C  PROMPT DESIGN FOR BENCHMARK CONSTRUCTION

To systematically construct our benchmark, we designed a set of task-oriented prompts that guide the generation of evaluation questions based on the information present in images. For each task scenario, the prompts explicitly define the core objectives, requirement categories, and output format to ensure consistency and rigor in the generated data. For instance, in the *Single-Answer* scenario, the prompt instructs the model to select one product as the correct answer while providing must-have and nice-to-have requirements with explicit priority levels. Similarly, for *Unanswerable* scenarios, the prompt is carefully constructed to ensure that no product in the image can satisfy all must-have requirements, while optional requirements create a controlled level of confusion.

**Prompt for Generating Requirements in Single-Answer Tasks for FTF-BENCH**

Given an image displaying products or services, generate a question that asks users to select a product or service based on their needs. Both the question and its answer must be strictly grounded in the information presented in the image; no content should be fabricated. The requirements are as follows:

**1. Determining the correct answer**
- Before generating the question, select one product or service from the candidates in the image as the correct answer.
- The question must be designed around this selected item, while one or more other items serve as distracting/confusing options.
- All content must be strictly based on the image; no invented details are allowed.
- **Core principle**: The question should test the ability to choose the correct item based on user requirements using the visual and contextual information in the image.

**2. Requirement categories and design**
- **Must-have requirements**:
- These requirements must uniquely identify the correct answer, forming a minimal necessary set.
- Each requirement in this set is indispensable—removing any single one may result in multiple items satisfying the criteria.
- The set must not contain redundant requirements—if removing a requirement still uniquely identifies the correct answer, that requirement is considered redundant and should be omitted.
- **Nice-to-have requirements**:
- These are optional requirements that may or may not be satisfied.
- They must be ordered by priority from highest to lowest.
- High-priority nice-to-have requirements are intended to make confusing options more appealing, while the correct answer must satisfy all must-have requirements and may satisfy only a subset of nice-to-have requirements.

**3. Design for confusion**
- The question should be challenging and complex.
- Distractor options should satisfy more of the high-priority nice-to-have requirements, whereas the correct answer fully satisfies must-have requirements but only partially satisfies nice-to-have requirements.
- This encourages users to overemphasize nice-to-have requirements and potentially overlook the must-have requirements, which are the decisive criteria.

**4. Output format**
- The output should be a JSON object with the following fields:
- `"Answer"`: the correct answer
- `"confusing_answer"`: the distracting option(s)
- `"must_have_requirements"`: a list of must-have requirements
- `"nice_to_have_requirements"`: a list of nice-to-have requirements ordered from highest to lowest priority

**Prompt for Generating Requirements in Multiple-Answer Tasks for FTF-BENCH**

Based on the products or services shown in the provided image, generate a question that asks users to select a product or service according to specific requirements. Both the question and the answer must be strictly grounded in the information visible in the image; no content should be fabricated. The requirements are as follows:

1. **Determining the correct answer**
- Before generating the question, select one product or service from the image as the correct answer.
- All questions must be designed around this selected product, while one or more other products serve as distracting/confusing options.
- Core principle: The question should reflect how the correct answer satisfies the requirements relative to the other candidates in the image.

2. **Requirement categories and design**
- **Must-have requirements**:
- These requirements should form a candidate pool containing multiple products from the image, including the correct answer.
- **Nice-to-have requirements**:
- Organize these into a list with explicit priority, from highest to lowest (smaller priority value indicates higher importance).
- By sequentially matching according to priority, the correct answer should be uniquely determined from the candidate pool.
- Requirement content should be realistic and contextually appropriate.

3. **Difficulty and distractor design**
- The question should be challenging, with complex and diverse requirements.
- High-priority nice-to-have requirements may conflict with must-have requirements to increase difficulty.
- Distractor options should satisfy more of the high-priority nice-to-have requirements but fail to meet must-have requirements; the correct answer may satisfy fewer high-priority nice-to-have requirements but must fully satisfy all must-have requirements.

4. **Output format**
- The output must be strictly in JSON format and contain only the JSON object, with the following fields:
- `"mandatory_requirements"`: a list of must-have requirements
- `"optional_requirements"`: a list of nice-to-have requirements with priorities, each represented as an object containing `"priority"` and `"requirement"`
- `"confusing_answer"`: the distracting option
- `"final_selected_product"`: the product ultimately selected after applying the optional requirements

Strictly follow the above rules and base the selection solely on the information provided in the task and visible in the image.

---

**Prompt for Generating Requirements in Unanswerable Tasks for FTF-BENCH**

Based on the products or services shown in the provided image, generate a question that asks users to select a product or service according to specific requirements. The question must be designed so that no product or service in the image can satisfy all core requirements. Both the question and its answer must be strictly grounded in the information visible in the image; no content should be fabricated. The requirements are as follows:

1. **Core objective**
- The generated question must ensure that all products or services in the image cannot become the correct answer; that is, no product/service can fully satisfy the task's core requirements.

2. **Requirement categories and design**
- **Must-have requirements**:
- Must satisfy two conditions:
1. Each individual requirement should be satisfied by at least one product or service in the image.
2. When all must-have requirements are combined, no single product or service should satisfy them all (i.e., each requirement has a corresponding "matching product," but no product matches all requirements).
- **Nice-to-have requirements (optional/distractor requirements)**:
- Organize as a list with explicit priority, from highest to lowest (smaller priority value indicates higher importance).
- Some optional requirements should be satisfied by most candidate products to create the illusion that a perfect product exists.
- Each requirement should be fluent, reasonable, and reflect common, real-world user needs.

3. **Difficulty and distractor design**
- The question should be complex and challenging, emphasizing confusion.
- Each product/service in the image should satisfy most of the must-have requirements, failing only on 1–2 critical requirements.
- Each product/service should satisfy some high-priority nice-to-have requirements, creating a "each has advantages" scenario, making it difficult to realize that no product fully satisfies all must-have requirements.

4. **Output format**
- The output must be strictly in JSON format and contain only the JSON object, with the following fields:
- `"mandatory_requirements"`: list of selected must-have requirements
- `"optional_requirements"`: list of selected nice-to-have requirements with priority, each represented as an object containing `"priority"` and `"requirement"`
- `"candidate_products"`: list of products that satisfy some, but not all, must-have requirements (serving as candidate options)

Strictly follow the above rules and base the selection solely on the information provided in the task and visible in the image. Ensure that no single product fully satisfies all must-have requirements.

---

## D    DETAILS OF HUMAN VERIFICATION

To ensure the quality and reliability of our dataset, we implemented a rigorous human verification process. Specifically, we developed a custom annotation script that streamlined the verification workflow, as shown in figure 3, allowing annotators to efficiently inspect each instance. The script presents the input data, associated candidate options, and relevant contextual information in a user-friendly interface, while logging the annotators' selections and comments for further analysis.

Annotators were instructed to evaluate multiple aspects of each item. These include the reasonableness of the question, the accuracy of the answer, the correctness of requirement categorization and requirement descriptions. Each instance was independently reviewed by at least two annotators. Disagreements were resolved through discussion or adjudication by a senior annotator. This procedure not only guarantees high annotation accuracy but also establishes a transparent and reproducible verification pipeline.

During the human verification process, we systematically examined the model's erroneous predictions and identified several prominent error types, as shown in Figure 4. The model often failed to follow *must-have requirements*, and sometimes treated *nice-to-have requirements* as mandatory and

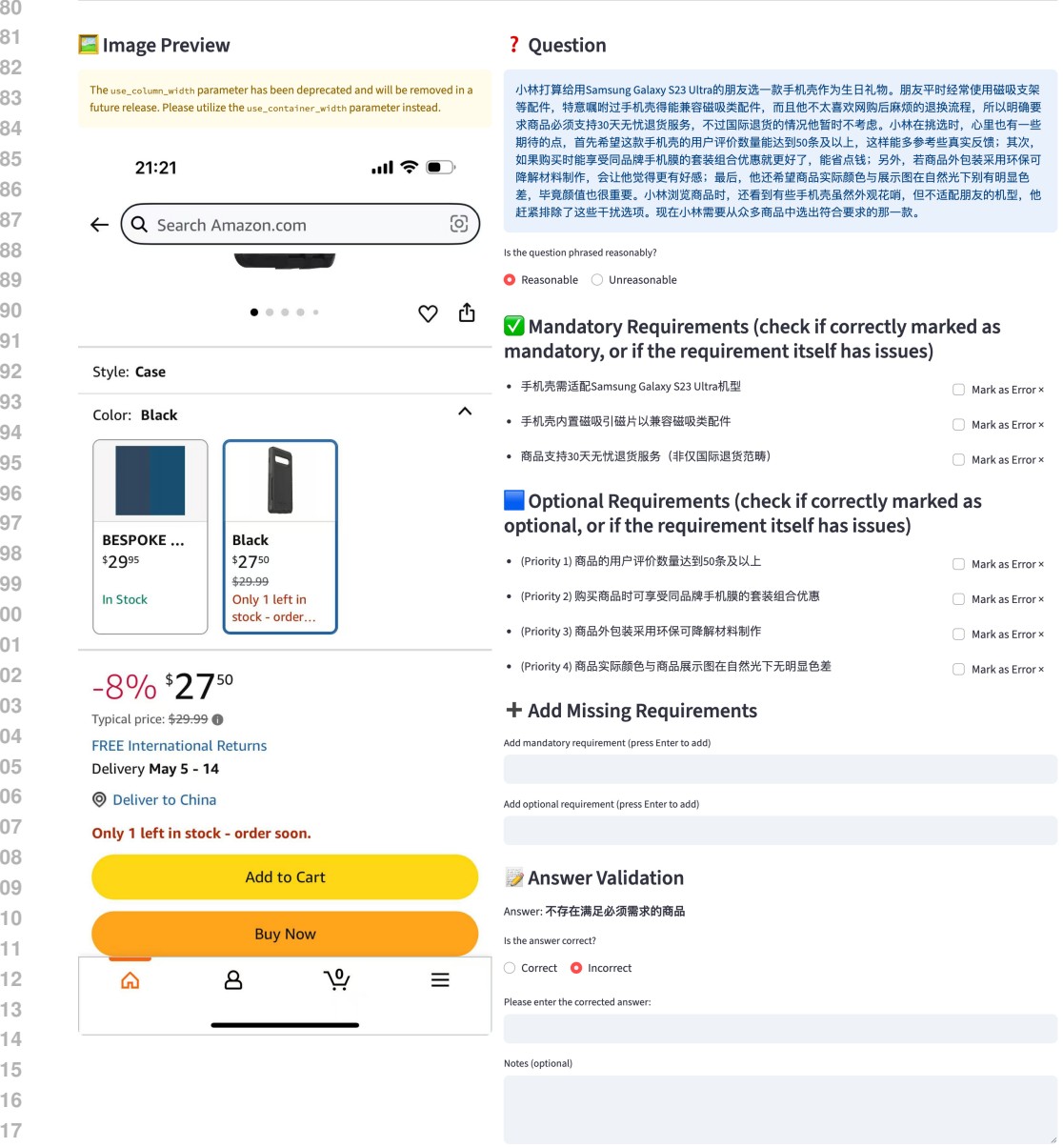

Figure 3: Screenshot of the human verification annotation interface. Annotators are asked to evaluate question reasonableness, answer accuracy, requirement categorization, and description correctness.

occasionally misordered them, reflecting challenges in understanding and prioritizing user preferences.

## E   ABLATION STUDY OF REWARD FUNCTIONS

To evaluate the contribution of requirement classification supervision, we conduct an ablation study by removing $R_{\text{requirement}}, R_{\text{format}}, R_{\text{answer}}$ from the overall objective and retraining the model under identical settings, respectively. The results across three benchmarks are summarized in Table 6. We observe a consistent performance drop across all datasets. The performance drop indicates that all reward components contribute to the final performance. This finding confirms that multi-objective reinforcement learning, which jointly optimizes for format compliance, answer accuracy,

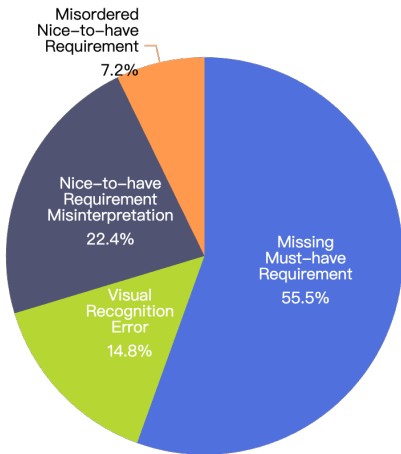

Figure 4: Distribution of model error types identified during human verification. Models frequently failed to follow *must-have requirements*, sometimes treated *nice-to-have requirements* as mandatory, and occasionally misordered them, highlighting challenges in understanding and prioritizing user preferences.

Table 6: Ablation study of the different rewards on Qwen2.5-VL-7B-Instruct.

| Setting | FTF-BENCH$_\text{val}$ | LogicVista | MathVision | InfoQA |
|---|---|---|---|---|
| **Full FTF-RL** | 55.8 | 47.4 | 25.7 | 67.8 |
| **Ablated (w/o $R_\text{requirement}$)** | 52.4↓3.4 | 44.7↓2.7 | 24.7↓1.0 | 60.3↓7.5 |
| **Ablated (w/o $R_\text{format}$)** | 54.5↓1.3 | 45.8↓1.6 | 25.1↓0.6 | 64.5↓3.3 |
| **Ablated (w/o $R_\text{answer}$)** | 48.3↓7.5 | 43.7↓3.7 | 24.5↓1.2 | 61.2↓6.6 |

requirement understanding, and reasoning quality, is crucial for improving accuracy in multimodal document understanding tasks.

## F   MODEL-DEPENDENT BIAS IN DATA CREATION AND EVALUATION

In our pipeline, we use `Doubao-Seed-1.6-250615` to generate the initial requirement sets and candidate answers, and also as the judger. All items are then checked by human annotators, who verify both the realism of the user requirements and the correctness of the final answer. Table 7 summarizes the joint distribution of human judgments on question reasonableness and answer correctness.

Most discrepancies come from prompts that annotators consider unrealistic in real life, while the majority of retained items have both reasonable requirements and correct answers. This indicates that `Doubao-Seed-1.6-250615` is capable of serving as a generator/judger, and that the subsequent human pass substantially reduces residual errors and potential same-source bias in the benchmark.

To further address the concern regarding model-dependent bias, we also substituted the judger model with GPT-5 while keeping the same evaluation protocol in Table 8. We observe very high cross-model agreement across all three scenarios, which suggests that label reliability does not depend on a single model family.

## G   REQUIREMENT CLASSIFICATION CONFUSION MATRICES

To better understand how MLLMs distinguish between must-have and nice-to-have requirements, we further evaluate their performance on a binary requirement-classification task over FTF-BENCH. For each requirement, we compare the gold label (must-have and nice-to-have) with the label predicted by the MLLM in the *Direct* setting and aggregate the counts into a confusion matrix.

Table 7: Joint distribution of human judgments on question reasonableness (question_reasonable) and answer correctness (has_error).

| Human judgment | Answer has error | Answer correct | Total |
|---|---|---|---|
| Question unreasonable | 683 (18.72%) | 51 (1.41%) | 734 (20.13%) |
| Question reasonable | 361 (9.90%) | 2554 (69.98%) | 2915 (79.87%) |
| Total | 1044 (28.62%) | 2605 (71.38%) | 3649 (100%) |

Table 8: Agreement between Doubao-Seed-1.6-250615 and GPT-5.

| Setting | Total | Num of Agree | Num of Disagree | Agree Rate |
|---|---|---|---|---|
| Single Answer | 3111 | 3093 | 17 | 99.42% |
| Multiple Answer | 3072 | 3057 | 14 | 99.51% |
| Unanswerable | 4764 | 4749 | 15 | 99.69% |

Table 9: Confusion matrix for requirement classification on FTF-BENCH (Direct setting) using Doubao-1.6-seed.

| Ground truth / Prediction | Predicted must-have | Predicted nice-to-have | Total |
|---|---|---|---|
| Must-have | 5221 (64.74%) | 2844 (35.26%) | 8065 |
| Nice-to-have | 2068 (16.79%) | 10246 (83.21%) | 12314 |
| Total | 7289 | 13090 | 20379 |

Table 10: Confusion matrix for requirement classification on FTF-BENCH (Direct setting) using GPT-5.

| Ground truth / Prediction | Predicted must-have | Predicted nice-to-have | Total |
|---|---|---|---|
| Must-have | 6425 (79.67%) | 1640 (20.33%) | 8065 |
| Nice-to-have | 1418 (11.52%) | 10896 (88.48%) | 12314 |
| Total | 7843 | 12486 | 20379 |

Table 11: Confusion matrix for requirement classification on FTF-BENCH (Direct setting) using Qwen2.5-VL-7B-Instruct.

| Ground truth / Prediction | Predicted must-have | Predicted nice-to-have | Total |
|---|---|---|---|
| Must-have | 4073 (50.52%) | 3992 (49.48%) | 8065 |
| Nice-to-have | 4229 (34.34%) | 8085 (65.66%) | 12314 |
| Total | 7843 | 12486 | 20379 |

Tables 9, 10, and 11 summarize the confusion matrices for Doubao-1.6-seed, GPT-5, and Qwen2.5-VL-7B-Instruct.

It can be seen that GPT-5 behaves in a relatively balanced way for both must-have and nice-to-have requirements. Doubao-1.6-seed shows strong recognition of nice-to-have requirements, but it also tends to classify more must-have requirements as nice-to-have. Qwen2.5-VL-7B-Instruct, as a representative open-source LLM, has a generally weaker capability to correctly separate the two types of requirements.

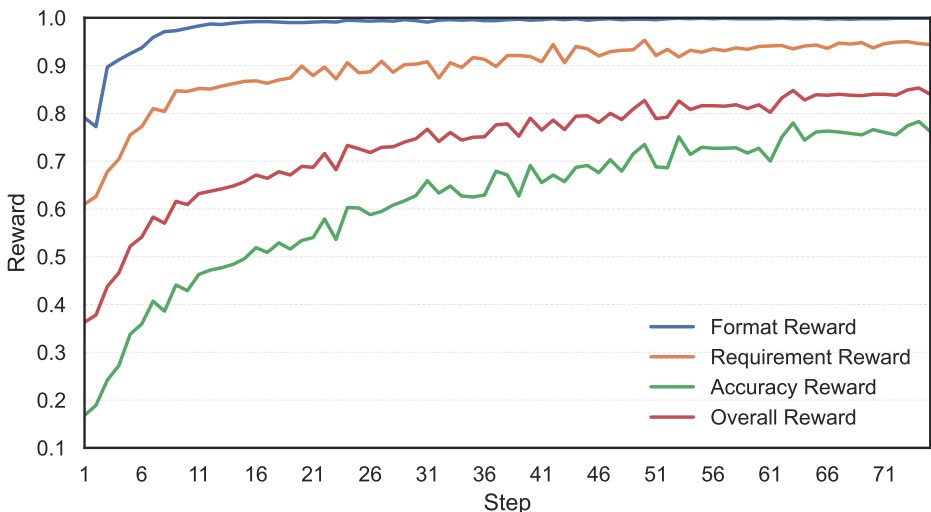

Figure 5: Training reward curves of Qwen2.5-VL-7B-Instruct under FTF-RL.

## H  TRAINING DYNAMICS OF FTF-RL ON QWEN2.5-VL-7B-INSTRUCT

We further analyze the behavior of FTF-RL by tracking the evolution of different rewards over the whole training process on Qwen2.5-VL-7B-Instruct. We monitor the format reward, the requirement classification reward, the answer correctness reward, and the aggregated overall reward at multiple training steps.

As shown in Figure 5, all four rewards improve monotonically during training, leading to a steady rise in the overall reward.

## I  QUALITATIVE CASE STUDIES ON GENERAL REASONING BENCHMARKS

To illustrate how requirement-aware reasoning learned from FTF-BENCH transfers to other tasks, we present two short case studies on LogicVista and MathVision using Qwen2.5-VL-7B-Instruct trained with FTF-RL. In both examples, the model only sees the original benchmark input and no additional supervision.

We provide the two cases in Table 12 and 13. In both cases, the trained model follows a consistent pattern. The first case is a MathVision geometry problem about a circular carpet on a tiled floor. The model must decide which grey tile pattern cannot come from any circle. After FTF-RL, Qwen2.5-VL-7B-Instruct does not jump directly to an option. It first summarizes the task requirement that grey tiles must be exactly those intersected by a single convex circle, so they should form one connected region with a smooth boundary. It then explicitly plans to check each candidate against this requirement and finally concludes that the option with a disconnected grey region is impossible. This shows that the model uses a requirement-driven plan rather than local pattern matching.

The second case is a LogicVista problem that involves inferring the meaning of two symbolic operations applied to shapes. The model must fill in a missing output shape and a missing operation symbol. After FTF-RL, Qwen2.5-VL-7B-Instruct begins by restating the subgoals, then applies these inferred rules to the two queries. It then follows this plan step by step and correctly selects the option. The model learns to organize the task into requirement extraction and execution.

## J  ADDITIONAL EXPERIMENTS ON COMPLEX AGENT TASKS

To further assess the generality of requirement-aware reasoning beyond the three service-oriented domains in our main experiments, we conduct additional evaluations on complex agent tasks. We select two representative benchmarks.

Table 12: Qualitative case study on MathVision after training on Qwen2.5-VL-7B-Instruct.

**Question.** A circular carpet is placed on a floor which is covered by equally big, square tiles. All tiles that have at least one point in common with the carpet are coloured in grey. Which of the following cannot be a result of this? <image>

**Image.**

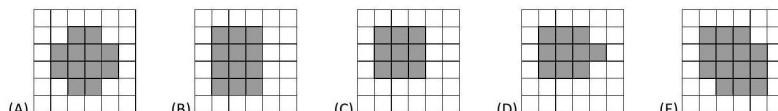

**Output.**
<requirements>"mandatory": ["Identify which grey tile pattern CANNOT result from coloring all tiles touching a circular carpet (tiles with at least one point in common with the circle)"],"optional": ["Explain why the pattern is invalid"]</requirements>
<think>
To solve this problem, we need to understand how placing a circular carpet would affect the surrounding squares if they touch it anywhere.
A circle is a convex, smooth, and symmetric shape. Tiles colored by a circle (all tiles intersecting the circle) must form a connected, "rounded" boundary (no sharp corners or disconnected "extensions" that cannot be touched by a circle).
Let's analyze each option:
A: The shape is connected with smooth edges. A circle could touch these tiles.
B: A tall, narrow shape—still connected and smooth; a circle could align with this.
C: A compact, connected shape; a circle could fit this.
D: The shape has a small extra tile on the right side but remains connected and smooth. A circle could touch this.
E: The the bottom-right tiles are separated from the main block by an empty tile. A circle is convex and continuous—it cannot touch tiles that are disconnected from the main group. The empty tile between the main block and the lower-right tiles means the circle cannot reach both regions without also touching the empty tile.
Thus, E cannot be the result.
</think>
<answer>E</answer>

AndroidControl (Li et al., 2024) evaluates mobile GUI control in realistic Android environments. It consists of human demonstrations of everyday tasks across diverse apps, where each trajectory records the screen observations, natural language instructions, and the corresponding low-level actions executed by the user. We evaluate performance on AndroidControl using Task Match Rate (TMR) and Action Match Rate (AMR) as our main metrics.

ScienceQA (Lu et al., 2022) captures multi-step scientific reasoning, which serves as a comprehensive testbed for multimodal reasoning across diverse science topics. It features a diverse collection of science questions covering 26 topics and 127 categories, where each example is annotated with CoT explanations and lectures to assess the model's ability to perform complex scientific reasoning and explanation generation. The metric is accuracy, measuring the number of corrected answers provided by models.

Table 14 and 15 report the performance before and after FTF-RL training on ScienceQA and AndroidControl, respectively. Across both benchmarks, we observe consistent improvements after FTF-RL, mirroring the gains we reported previously on LogicVista, MathVision, and MathVista. This shows that learning to prioritize requirements on FTF-BENCH transfers to more complex agent behaviors.

Table 13: Qualitative case study on LogicVista after training on Qwen2.5-VL-7B-Instruct.

**Question.** Based on the diagram at the top of the page, which two of the proposals A, B, C or D completes the diagrams at the bottom of the page? <image>

**Image.**

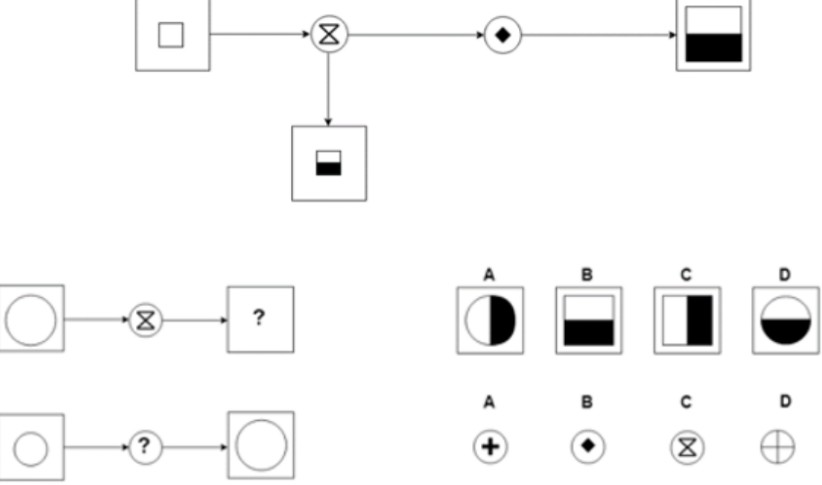

**Output.**

<requirements>"mandatory": ["Analyze the top diagram to determine the transformation rules of operations for shapes within square frames","Apply the rules to select the correct shape (from A/B/C/D) for the '?' in the first bottom diagram","Apply the rules to select the correct operation (from A/B/C/D) for the '?' in the second bottom diagram"],"optional": ["Explain each operation's function with reference to the top diagram"]</requirements>

<think>

First, I need to re-extract the operation rules from the top diagram. The top diagram has two inputs: a square with a small white square, and a square with a small square.

Operation cross symbol combines the "small inner shapes" of the two inputs.

Operation the diamond symbol then scales the combined small shape to fill the entire outer square frame.

Next, for the first bottom diagram: Input is a square with a small white circle.

Operation cross symbol should combine this small circle with the "bottom half black" fill.

So the result should be a square frame containing a small circle (bottom half black) — which is option D.

For the second bottom diagram:

Input is a square with a small white circle; output is a square with a large circle.

This scaling function matches operation the diamond symbol from the top diagram — which is option B.

</think>

<answer>D, B</answer>

Table 14: Performance on ScienceQA after training with FTF-RL on FTF-BENCH.

| Model | ScienceQA |
|---|---|
| Qwen2.5-VL-7B-Instruct | 40.06 |
| +FTF-RL | **40.28**↑0.22 |

Table 15: Performance on AndroidControl after training with FTF-RL on FTF-BENCH.

| Model | Metric | CLICK | TYPE | SCROLL | OPENAPP | WAIT | COMPLETE | PRESS |
|---|---|---|---|---|---|---|---|---|
| Qwen2.5-VL-7B-Instruct | TMR | 0.9538 | 0.8880 | 0.8571 | 0.0000 | 0.0459 | 0.9339 | 0.2857 |
| | AMR | 0.3317 | 0.6800 | 0.0603 | 0.0000 | 0.0459 | 0.9339 | 0.2857 |
| +FTF-RL | TMR | 0.9608↑0.0071 | 0.9216↑0.0336 | 0.8026↓0.0545 | 0.0000↑0.0000 | 0.0529↑0.0071 | 0.9533↑0.0194 | 0.5364↑0.2507 |
| | AMR | 0.3240↓0.0077 | 0.7184↑0.0384 | 0.0446↓0.0157 | 0.0000↑0.0000 | 0.0529↑0.0071 | 0.9533↑0.0194 | 0.5364↑0.2507 |

Table 16: Performance of all models on the full FTF-BENCH and on the 10% evaluation subset.

| Model | Split | Sin. Upper | Sin. Direct | Mul. Upper | Mul. Direct | Unans. Upper | Unans. Direct | Avg. Upper | Avg. Direct |
|---|---|---|---|---|---|---|---|---|---|
| Gemini-2.5-pro | Full | 88.89 | 86.91 | 84.20 | 82.26 | 81.72 | 78.55 | 84.26 | 81.75 |
| | Subset | 79.61 | 81.55 | 72.28 | 69.31 | 63.29 | 58.23 | 69.16 | 66.57 |
| GPT-o3 | Full | 82.67 | 77.78 | 80.41 | 80.03 | 83.31 | 83.09 | 82.33 | 79.68 |
| | Subset | 75.73 | 72.15 | 69.31 | 75.25 | 81.01 | 73.79 | 75.22 | 72.33 |
| Doubao-1.6-seed | Full | 80.09 | 78.66 | 81.70 | 76.99 | 83.80 | 84.05 | 82.24 | 80.59 |
| | Subset | 72.82 | 66.99 | 63.37 | 58.42 | 74.68 | 68.99 | 69.74 | 63.98 |
| LLaMA-4 | Full | 58.36 | 55.48 | 55.93 | 54.45 | 44.43 | 36.94 | 55.03 | 51.99 |
| | Subset | 54.37 | 49.51 | 41.58 | 25.32 | 30.38 | 40.59 | 37.75 | 33.72 |
| Qwen2.5-VL-7B-Instruct | Full | 61.16 | 22.99 | 58.25 | 20.10 | 47.59 | 18.64 | 57.69 | 21.05 |
| | Subset | 57.28 | 20.39 | 39.60 | 11.88 | 37.34 | 9.49 | 41.21 | 9.51 |
| Qwen2.5-VL-32B-Instruct | Full | 70.66 | 69.25 | 70.55 | 70.17 | 53.08 | 50.58 | 67.72 | 66.57 |
| | Subset | 66.99 | 67.96 | 42.41 | 34.18 | 54.46 | 58.42 | 50.72 | 48.41 |

Table 17: DOUBAO-1.6-SEED accuracy on FTF-BENCH under different decoding temperatures.

| Temperature | Single-Answer | Multiple-Answer | Unanswerable | Average |
|---|---|---|---|---|
| 0.0 | 78.66 | 76.99 | 84.05 | 80.59 |
| 0.2 | 88.16 | 80.57 | 73.61 | 81.70 |
| 0.4 | 79.41 | 80.96 | 62.28 | 72.21 |
| 0.6 | 81.10 | 79.10 | 61.37 | 71.92 |
| 0.8 | 90.24 | 82.89 | 77.55 | 82.65 |
| 1.0 | 89.47 | 82.29 | 77.81 | 82.38 |

## K    ANALYSIS OF THE 10% EVALUATION SUBSET

In the RL experiments, we randomly sample 90% of FTF-Bench for training and hold out the remaining 10% for evaluation, and the same subset is used for all models. To check whether this subset introduces bias, we evaluate all baseline models on both the full benchmark and this 10% subset and compare the results in Table 16.

## L    ADDITIONAL ANALYSIS ON DECODING TEMPERATURE

We ran an additional experiment on DOUBAO-1.6-SEED with different decoding temperatures, including a fully deterministic setting with temperature 0.0. Table 17 reports the accuracy on FTF-BENCH for temperatures between 0.0 and 1.0.

The overall accuracy varies only moderately across temperatures, indicating that the tendency to refuse is stable.

We further measure the agreement between outputs at temperature 0 and outputs at other temperatures on an instance-by-instance basis. Results are shown in Table 18.

This high consistency shows that the refusal behavior and the error patterns we analyze are stable. We have included the results above in Appendix L of the revised version of our paper.

Table 18: Instance-level agreement between temperature $0.0$ and higher temperatures for DOUBAO-1.6-SEED.

| Temperature | Single-Answer | Multiple-Answer | Unanswerable | Average |
|---|---|---|---|---|
| 0.2 | 79.19 | 75.24 | 80.89 | 77.79 |
| 0.4 | 74.00 | 73.33 | 67.16 | 70.72 |
| 0.6 | 73.55 | 73.47 | 66.55 | 70.47 |
| 0.8 | 79.11 | 75.22 | 78.34 | 77.74 |
| 1.0 | 78.84 | 75.54 | 79.21 | 78.13 |

