# OpenReview forum: "First Things First: Teaching LLM-Based Agents to Prioritize Must-Haves before Nice-to-Haves"
_ICLR.cc/2026/Conference — Submitted to ICLR 2026_

### Official Review · Reviewer_SuaU · 2025-10-26

**Soundness:** 3
**Presentation:** 3
**Contribution:** 3
**Rating:** 6
**Confidence:** 4

**Summary:**

The paper tackles the problem that current Multimodal Large Language Models (MLLMs), when acting as agents, often fail to handle complex user requests that include both essential "must-have" requirements and desirable "nice-to-have" preferences. Existing models tend to either violate must-haves or fail when requirements conflict. To address this, the authors introduce: FTF-BENCH: A new benchmark with 3,649 problems across realistic domains (e-commerce, booking, maps) designed to test MLLMs' ability to prioritize requirements.

**Strengths:**

1. The paper highlights a critical flaw in current agents, their inability to handle requirement priorities, which is crucial for real-world usability.

2. The benchmark is comprehensive, covers realistic domains, and crucially includes the three distinct answer scenarios (Single, Multiple, Unanswerable) needed to properly evaluate requirement prioritization.

3. The paper demonstrates catastrophic failures in existing models and significant, consistent improvements with FTF-RL.

**Weaknesses:**

1. The benchmark uses synthesized user requests. Real user requests can be far more ambiguous, implicit, or contradictory than the structured (even if colloquial) prompts generated for the benchmark.

2.  Implementing a multi-objective RL framework like FTF-RL is significantly more complex than standard supervised fine-tuning (SFT) or basic RLHF, potentially limiting its adoption.

3. While the domains (shopping, booking, maps) are relevant, the findings might not directly generalize to all types of agent tasks (e.g., complex software control, scientific discovery).

**Questions:**

1. How robust is FTF-RL when faced with genuinely ambiguous user requests where the distinction between a must-have and a strong preference is unclear even to a human?

2. FTF-RL uses a rule-based reward model. How well would this scale to more complex tasks with potentially hundreds of implicit or explicit requirements? Would it require excessive manual effort to define rewards?

3.  Could models trained with FTF-RL become overly rigid, always asking for explicit must-haves vs. nice-to-haves, potentially disrupting natural conversation flow in simpler scenarios?

---

> ### Author Response · Authors · 2025-11-23
> **Response to Reviewer SuaU (Part 1)**
>
> We thank the reviewer for the thoughtful feedback. We address the comments below.
>
> * * *
> > **Weaknesses 1**: The benchmark uses synthesized user requests. Real user requests can be far more ambiguous, implicit, or contradictory than the structured (even if colloquial) prompts generated for the benchmark.
>
> Actually, our goal is to generate realistic, ambiguous, and contradictory instructions. Specifically, we explicitly prompt Doubao-Seed-1.6-250615 as the generator to generate ambiguous and colloquial descriptions that approximate complex user requirements rather than cleanly enumerated conditions. These synthesized requests are then checked by human annotators, who filter out unnatural or overly template-like cases and retain only those that remain colloquial while still grounding a well-defined must-have and nice-to-have structure.
>
> In addition, we agree that AI-generated queries cannot fully capture the richness of real-world human requests and that real users can express their needs in even more abstract ways. In our view, **this limitation actually makes our empirical findings conservative**. Even under the current human-verified synthetic setting, we already see large gaps between the Direct and Upper conditions and catastrophic failures. More complex real-world requests would likely further enlarge the gap between raw model behavior and the upper bound with gold requirements, reinforcing rather than weakening our conclusion that better requirement understanding is urgently needed.
>
> * * *
> > **Weaknesses 2**: Implementing a multi-objective RL framework like FTF-RL is significantly more complex than standard supervised fine-tuning (SFT) or basic RLHF, potentially limiting its adoption.
>
> Our proposed training method is simple and flexible. **FTF-RL is built on a standard RL pipeline and introduces only a rule-based multi-objective reward without training any additional reward model or collecting new human preference data.** In practice, the main extra engineering effort is to parse the XML-style \<requirements\>, \<think\>, and \<answer\> fields, which can be integrated into existing RLHF infrastructure with minimal modification.
>
> More importantly, the main contribution of this paper is not to propose another general RL method, but to investigate how explicitly modeling must-have and nice-to-have requirements affects the general reasoning capabilities of MLLMs. We view FTF-RL as one concrete instantiation of this idea. Future work can further simplify the training pipeline, but our results demonstrate that incorporating requirement-aware signals into RL is technically feasible and empirically efficient.
>
> * * *
> > **Weaknesses 3**: While the domains (shopping, booking, maps) are relevant, the findings might not directly generalize to all types of agent tasks (e.g., complex software control, scientific discovery).
>
> We thank the reviewer for pointing out the concern about generalization beyond shopping, booking, and map scenarios. We agree that these domains cover only a subset of real agent tasks. However, our core contribution is not domain specific templates but a requirement-aware formulation and RL method that explicitly teaches models to prioritize requirements and learn to plan. This is agnostic to the surface domain and applies whenever an agent must satisfy multiple complex requirements.
>
> We have added new experiments in Appendix J of the revised version. We evaluate models fine tuned with FTF-RL on two additional public benchmarks. AndroidControl represents complex software control in an operating system environment. ScienceQA captures multi-step scientific reasoning.
>
> Across both benchmarks we observe consistent improvements after FTF-RL, mirroring the gains we reported previously on LogicVista, MathVision, and MathVista. This shows that learning to prioritize requirements on FTF-Bench transfers to more complex agent behaviors.
>
> | Model | $\text{ScienceQA}$ |
> |-|-|
> | Qwen2.5 VL 7B Instruct | 40.06 |
> | $\quad$+FTF-RL | $40.28_{\uparrow 0.22}$ |
>
> | Model | Metric | CLICK | TYPE | SCROLL | OPENAPP | WAIT | COMPLETE | PRESS |
> |-|-|-|-|-|-|-|-|-|
> | Qwen2.5-VL-7B-Instruct | TMR | 0.9538 | 0.8880 | 0.8571 | 0.0000 | 0.0459 | 0.9339 | 0.2857 |
> | Qwen2.5-VL-7B-Instruct | AMR | 0.3317 | 0.6800 | 0.0603 | 0.0000 | 0.0459 | 0.9339 | 0.2857 |
> | $\quad$+FTF-RL | TMR | $0.9608_{\uparrow 0.0071}$ | $0.9216_{\uparrow 0.0336}$ | $0.8026_{\downarrow 0.0545}$ | $0.000_{\uparrow 0.0000}$ | $0.0529_{\uparrow 0.0071}$ | $0.9533_{\uparrow 0.0194}$ | $0.5364_{\uparrow 0.2507}$ |
> | $\quad$+FTF-RL | AMR | $0.3240_{\downarrow 0.0077}$ | $0.7184_{\uparrow 0.0384}$ | $0.0446_{\downarrow 0.0157}$ | $0.000_{\uparrow 0.0000}$ | $0.0529_{\uparrow 0.0071}$ | $0.9533_{\uparrow 0.0194}$ | $0.5364_{\uparrow 0.2507}$ |

---

> ### Author Response · Authors · 2025-11-23
> **Response to Reviewer SuaU (Part 2)**
>
> * * *
> > **Questions 1**: How robust is FTF-RL when faced with genuinely ambiguous user requests where the distinction between a must-have and a strong preference is unclear even to a human?
>
> Since this suggestion is similar to Weaknesses 1, we addressed the reviewer's concern about this point in our response to Weaknesses 1.
>
> We have also provided an example in Figure 2 that makes the boundary between must-have and nice-to-have ambiguous even for humans. FTF-RL requires the model to first produce an explicit JSON parse within the `<requirements>...</requirements>` block, where it must separate "mandatory" from "optional" items before continuing to think and answer. This explicit requirement-classification step makes the policy more robust under ambiguity.
>
> * * *
> > **Questions 2**: FTF-RL uses a rule-based reward model. How well would this scale to more complex tasks with potentially hundreds of implicit or explicit requirements? Would it require excessive manual effort to define rewards?
>
> Our work is primarily motivated by requirement-aware service scenarios such as e-commerce, booking, and map or ride-hailing interfaces, where a user query typically contains only a small number of distinct must-have and nice-to-have requirements, usually fewer than 10. FTF-Bench is constructed to mirror this reality rather than a synthetic setting with hundreds of specified verbal requirements, which we rarely observe in real user prompts.
>
> More importantly, training on tasks with a small number of explicit requirements does not mean that the improved models are only better at handling up to 10 requirements. We already evaluated trained models on external reasoning benchmarks. These benchmarks involve long multi-step reasoning and compositional constraints, yet FTF-RL still brings consistent gains. This suggests that optimizing requirement-aware reasoning induces broader improvements in planning beyond the exact training format of FTF-Bench.
>
> Finally, in response to Weakness 3, we have further added experiments on three substantially more complex generalization settings, covering complex software control, scientific discovery, and web information retrieval. These tasks require the model to integrate many implicit and explicit factors and balance a large number of potential requirements that cannot be enumerated as a short list. Even in these settings, we observe improvements after applying FTF-RL, indicating that the learned behavior scales to scenarios where the decision space depends on many more than a handful of requirements.
>
> * * *
> > **Questions 3**: Could models trained with FTF-RL become overly rigid, always asking for explicit must-haves vs. nice-to-haves, potentially disrupting natural conversation flow in simpler scenarios?
>
> We thank the reviewer for raising this concern about potential rigidity in models trained with FTF-RL. First, FTF-RL does not train the model to ask users to explicitly list must-haves and nice-to-haves. The model receives a single colloquial query that freely mixes different requirements, and it is rewarded for internally inferring the necessity hierarchy and then solving the task. The structured requirement JSON is only an internal supervision signal used during RL, not a user-facing interaction pattern. At deployment, one can omit the requirement tags entirely and simply use the model as a standard assistant, while still benefiting from the improved requirement-aware reasoning.
>
> Second, the RL objective is carefully regularized so that the updated policy stays close to the base model. We use a KL penalty toward the original instruction-tuned checkpoint, which explicitly discourages overly rigid behavioral shifts. Empirically, this is reflected in FTF-Bench single-answer scenarios that correspond to simpler queries without complicated trade-offs. On these easier cases, FTF-RL brings consistent gains rather than degradation for both 3B and 7B models, suggesting that the model learns to respect hard requirements better without becoming brittle in straightforward interactions.
>
> Third, if FTF-RL were making the model overly rigid, we would expect a drop in general reasoning ability once the model is optimized for requirement parsing. Instead, when evaluated on LogicVista, MathVision, and MathVista, models trained only on FTF-Bench with FTF-RL obtain consistent improvements over the same backbones. This shows that enhancing requirement awareness does not come at the cost of general reasoning. Rather, by teaching the model to explicitly identify hard requirements, filter candidates, and then reason within the viable set, FTF-RL indirectly strengthens its ability to structure requirements in diverse reasoning tasks.
>
> * * *
> We hope that our explanations above can clarify your doubts and you can consider our work more favorably.

---

> > ### Comment · Reviewer_SuaU · 2025-11-26
> >
> > Thanks the authors for their rebuttal. After reviewing other reviewers' comments and careful consideration, I decide to remain my score.

---

> > > ### Author Response · Authors · 2025-11-27
> > >
> > > Thank you for acknowledging our response and your kind encouragement. Please do not hesitate to contact us if you have any further questions or require additional information.

---

### Official Review · Reviewer_Gh4d · 2025-10-28

**Soundness:** 2
**Presentation:** 3
**Contribution:** 2
**Rating:** 2
**Confidence:** 3

**Summary:**

This paper proposes the FTF-RL (First Things First Reinforcement Learning) approach, which optimizes model reasoning through a multi-objective reward function encompassing format compliance, answer correctness, and requirement classification accuracy. It has achieved good performance on the FTF-BENCH proposed in this paper and some public benchmarks.

**Strengths:**

1. This paper is clearly written, with explicit introductions to its methods and experiments, making it easy for readers to follow .
2. The research field focused on in this paper holds significant research value .
3. This paper constructs a new evaluation benchmark, which is conducive to the development of this field .

**Weaknesses:**

1. The comparative experiments in the paper are really weak. They only compare the performance before and after using FTF-RL, without any horizontal comparison with other strategies. This makes it impossible to demonstrate the relative advantages of the proposed method and is completely insufficient to support the paper's conclusions.
2. The ablation experiments are also incomplete, as they only demonstrate the role of the single module $R_{requirement}$.
3. The paper has limited innovation. Although the proposed FTF-BENCH covers the "must-have requirement - nice-to-have requirement" hierarchy, its core idea still does not deviate from the existing benchmark paradigm of "instruction following + multi-scenario verification", and the multi-objective reward function is more of a combination of existing technologies.

**Questions:**

None.

---

> ### Author Response · Authors · 2025-11-23
> **Response to Reviewer Gh4d (Part 1)**
>
> We thank the reviewer for the thoughtful feedback. We address the comments below.
>
> * * *
> > **Weaknesses 1**: The comparative experiments in the paper are really weak. They only compare the performance before and after using FTF-RL, without any horizontal comparison with other strategies. This makes it impossible to demonstrate the relative advantages of the proposed method and is completely insufficient to support the paper's conclusions.
>
> **To our knowledge, this paper is the first to explicitly investigate requirement-aware reasoning** for MLLMs under a necessity hierarchy of must-have and nice-to-have requirements. Existing instruction following baselines do not target this scenario and do not provide strategies that explicitly prioritize requirements, which makes a direct horizontal comparison with prior requirement-aware training algorithms impossible.
>
> Instead, our key insight is that requirement-aware reasoning is crucial for improving how MLLMs understand user intent and for strengthening their general reasoning ability, and **the main contribution of this work lies in establishing this task rather than proposing another training technique**. Table 2 of our paper evaluates a broad set of strong MLLMs on FTF-Bench, revealing that current instruction-tuned models all suffer from catastrophic failures in this setting, which establishes a strong baseline and emphasizes the necessity of requirement-aware reasoning.
>
> We further present a comparative experiment on LLaVA-OneVision-7B and LLaVA-1.5-13B. Both models have undergone various forms of fine-tuning on general-purpose corpora. Results show that they exhibit significant improvements not only in requirement-aware reasoning itself but also in reasoning tasks across other scenarios. We have included the results below in Table 3 and 4 of the revised version of our paper.
>
> Impact of reinforcement learning on LLaVA variants:
> | Models | Single-Answer | Multiple-Answer | Unanswerable | Average |
> |-|-|-|-|-|
> | LLaVA-OneVision-7B | 45.82 | 33.69 | 42.87 | 44.09 |
> | $\quad$+FTF-RL | **$52.51_{\uparrow 6.69}$** | **$40.15_{\uparrow 6.46}$** | **$48.79_{\uparrow 5.92}$** | **$50.48_{\uparrow 6.39}$** |
> | LLaVA-1.5-13B | 49.23 | 37.51 | 43.19 | 46.64 |
> | $\quad$+FTF-RL | **$57.81_{\uparrow 8.58}$** | **$43.74_{\uparrow 6.23}$** | **$49.05_{\uparrow 5.86}$** | **$52.87_{\uparrow 7.23}$** |
>
> Effect of FTF-RL across general reasoning benchmarks on LLaVA variants:
> | Models | LogicVista | MathVision | MathVista |
> |-|-|-|-|
> | LLaVA-1.5-13B | 29.23 | 11.12 | 27.64 |
> | $\quad$+FTF-RL | **$35.61_{\uparrow 6.38}$** | **$13.74_{\uparrow 2.62}$** | **$29.05_{\uparrow 1.41}$** |
>
> * * *
> > **Weaknesses 2**: The ablation experiments are also incomplete, as they only demonstrate the role of the single module $R_{requirement}$.
>
> We thank the reviewer for pointing out the concern about the ablation studies. Our main algorithmic novelty lies in the requirement reward $R_{requirement}$, which directly supervises the distinction between must-have and nice-to-have requirements and is therefore the component most tightly coupled with the central goal of FTF-RL. For this reason, our paper focuses first on the ablation of $R_{requirement}$ in Section 5.3.3. Due to space limitations, we have moved the ablation experiment to Appendix E.
>
> In addition, we have now extended the ablation studies to the other reward components of FTF-RL. Both ablations lead to a performance drop compared with the full FTF-RL model, which shows that all reward components contribute to the final performance. We have included the extended ablation studies in Appendix E of the revised version.
>
> | Setting | FTF-Bench | LogicVista | MathVision | InfoQA |
> |-|-|-|-|-|
> | Full FTF-RL | 55.8 | 47.4 |25.7 | 67.8 |
> | w/o $R_{requirement}$ | **$52.4_{\downarrow 3.4}$** | **$44.7_{\downarrow 2.7}$** | **$24.7_{\downarrow 1.0}$** | **$60.3_{\downarrow 7.5}$** |
> | w/o $R_{format}$ | **$54.5_{\downarrow 1.3}$** | **$45.8_{\downarrow 1.6}$** | **$25.1_{\downarrow 0.6}$** | **$64.5_{\downarrow 3.3}$** |
> | w/o $R_{answer}$ | **$48.3_{\downarrow 7.5}$** | **$43.7_{\downarrow 3.7}$** | **$24.5_{\downarrow 1.2}$** | **$61.2_{\downarrow 6.6}$** |

---

> ### Author Response · Authors · 2025-11-23
> **Response to Reviewer Gh4d (Part 2)**
>
> * * *
> > **Weaknesses 3**: The paper has limited innovation. Although the proposed FTF-BENCH covers the "must-have requirement - nice-to-have requirement" hierarchy, its core idea still does not deviate from the existing benchmark paradigm of "instruction following + multi-scenario verification", and the multi-objective reward function is more of a combination of existing technologies.
>
> **The main focus of FTF-Bench is requirement-aware reasoning rather than generic "instruction following + multi-scenario verification"**. Existing instruction-following benchmarks typically treat all requirements as equally important and mainly check whether the model complies with a flat set of requirements. In contrast, FTF-Bench explicitly introduces a requirement hierarchy over user requirements, with clearly separated must-have and nice-to-have requirements, and structures problems into Single-Answer, Multiple-Answer, and Unanswerable settings. To our knowledge, this requirement-priority perspective on multimodal service scenarios is not captured by existing benchmarks, even if they share surface similarities in being "instruction-following" and "multi-scenario".
>
> * * *
> We hope that our explanations above can clarify your doubts and you can consider our work more favorably.

---

### Official Review · Reviewer_Dr9c · 2025-11-01

**Soundness:** 3
**Presentation:** 2
**Contribution:** 2
**Rating:** 4
**Confidence:** 3

**Summary:**

The paper introduces a requirement hierarchy—must-have vs. nice-to-have—for real-world service-oriented multimodal agents. It builds FTF-BENCH (3,649 examples across e-commerce, ticketing/booking, and maps/transport) with three settings: single-solution, multi-solution, and unanswerable (requiring refusal). The authors propose FTF-RL, which adds three reward components in reinforcement learning—structured formatting, final-answer correctness, and requirement classification—to force the model to first parse and separate requirements before choosing and reasoning. Across multiple MLLMs, Direct is far below the Upper bound (given gold requirements); FTF-RL closes the gap substantially and shows some out-of-distribution gains on LogicVista/MathVision/MathVista.

**Strengths:**

**Novel angle**: In real services, the must-have vs. nice-to-have hierarchy is crucial yet often overlooked; FTF-BENCH quantifies this gap.

**Benchmark design addresses a blind spot**: The large gap between Direct and Upper helps disentangle perception errors from requirement parsing/priority errors.

**Transfer signal**: Gains on LogicVista/MathVision/MathVista suggest the “parse requirements first, then reason” paradigm has broader applicability.

**Weaknesses:**

**Task narrowness**: Although the angle is strong, the scenarios are relatively limited, making the practical generality of the approach less clear.

**Lack of tests on public benchmarks**: Because the task setup is narrow, the data distribution during testing is also narrow; it’s hard to judge whether the approach works across tasks in the same broad category (e.g., web information retrieval, etc.).

**Model-dependent bias in data creation and evaluation**: Prompts/requirements are model-generated and then human-checked; final-answer correctness is judged by a strong MLLM. The paper should disclose the specific generator/judger models and report human–model agreement, plus variance when swapping to a different model family, to reduce same-source bias and evaluation skew.

**Questions:**

**Judger and agreement**: Which model/version is used as the judger? What is the agreement with two human annotators on a random subset? What is the variance when replacing the judger with a different model family? (Please include sample size.)

**Unanswerable threshold and calibration**: What triggers refusal? Did you try confidence calibration or temperature scaling?

**Add at least one public benchmark**: Please include evaluation on at least one public benchmark.

---

> ### Author Response · Authors · 2025-11-23
> **Response to Reviewer Dr9c (Part 1)**
>
> We thank the reviewer for the thoughtful feedback. We address the comments below.
>
> * * *
> > **Weaknesses 1**: Task narrowness: Although the angle is strong, the scenarios are relatively limited, making the practical generality of the approach less clear.
>
> Our approach enjoys practical generality via the following aspects. Firstly, it is deliberately designed to cover a broad range of realistic situations where users express vague and composite requirements. The benchmark spans e-commerce, booking services such as hotels and transport, and map or ride-hailing interfaces, with screenshots taken from diverse interface states. Within these domains, we explicitly include the three key real-world cases of requirement-aware interaction with a unique feasible option, multiple feasible options that must be traded off, and infeasible requests that require abstention. These are precisely the types of scenarios that most frequently arise in practice rather than arbitrarily constructed or unrealistic cases.
>
> More importantly, our main goal is not to solve only these specific screen selection tasks, but to strengthen MLLMs in requirement-aware reasoning more broadly. Models optimized with FTF-RL are not only better on FTF-Bench itself, but they also improve consistently on four external reasoning benchmarks. We have also added experiments on three substantially more complex generalization settings that go beyond the original service interfaces in response to Weaknesses 2 below. These results show that training on FTF-Bench provides a broadly useful signal that enhances MLLM's ability to understand and plan under vague requirements beyond the specific training tasks.
>
> * * *
> > **Weaknesses 2**: Lack of tests on public benchmarks: Because the task setup is narrow, the data distribution during testing is also narrow; it’s hard to judge whether the approach works across tasks in the same broad category (e.g., web information retrieval, etc.).
>
> We thank the reviewer for pointing out the concern about the task diversity. We have added new experiments in Appendix J of the revised version. We evaluate models fine tuned with FTF-RL on the web information retrieval benchmark BRIGHT. , which is designed to evaluate reasoning-intensive retrieval capabilities in realistic scenarios. We evalate on the three subsets below. The metric is nDCG@10, which measures the ranking quality of the top 10 retrieved results, accounting for both the relevance and position of each item by comparing to an ideal ranking.
>
> Across all subsets, we observe consistent improvements after FTF-RL, mirroring the gains we reported previously on LogicVista, MathVision, and MathVista. This shows that learning to prioritize requirements on FTF-Bench transfers to more complex agent behaviors.
>
> | Model | $\text{BRIGHT}_\text{biology}$ | $\text{BRIGHT}_\text{pony}$ | $\text{BRIGHT}_\text{psychology}$ |
> |-|-|-|-|
> | Qwen2.5 VL 3B Instruct | 0.86 | 3.14 | 0.05 |
> | $\quad$+FTF-RL | **$1.11_{\uparrow 0.25}$** | **$6.80_{\uparrow 3.66}$** | **$0.07_{\uparrow 0.02}$** |

---

> ### Author Response · Authors · 2025-11-23
> **Response to Reviewer Dr9c (Part 2)**
>
> * * *
> > **Weaknesses 3**: Model-dependent bias in data creation and evaluation: Prompts/requirements are model-generated and then human-checked; final-answer correctness is judged by a strong MLLM. The paper should disclose the specific generator/judger models and report human–model agreement, plus variance when swapping to a different model family, to reduce same-source bias and evaluation skew.
>
> In our pipeline, we use Doubao-Seed-1.6-250615 to generate the initial requirement sets and candidate answers, and also as the judger. All items are then checked by human annotators, who verify both the realism of the user requirements and the correctness of the final answer. The table below summarizes the joint distribution of human judgments on question reasonableness and answer correctness. We have included the results below in Appendix F of the revised version of our paper.
>
> | Human judgment | Answer has error | Answer correct | Total |
> |-|-|-|-|
> | Question unreasonable  | 683 (18.72%) | 51 (1.41%) | 734 (20.13%) |
> | Question reasonable | 361 (9.90%) | 2554 (69.98%) | 2915 (79.87%) |
> | Total | 1044 (28.62%) | 2605 (71.38%) | 3649 (100%) |
>
> Most discrepancies come from prompts that annotators consider unrealistic in real life, while the majority of retained items have both reasonable requirements and correct answers. This indicates that Doubao-Seed-1.6-250615 is capable of serving as a generator/judger, and that the subsequent human pass substantially reduces residual errors and potential same-source bias in the benchmark.
>
> To further address the concern regarding model-dependent bias, we also substituted the judger model with GPT-5 while keeping the same evaluation protocol. We observe very high cross-model agreement across all three scenarios, which suggests that label reliability does not depend on a single model family. We have included the results below in Appendix F of the revised version of our paper.
>
> | Setting | Total | Num of Agree | Num of Disagree | Agree Rate |
> |-|-|-|-|-|
> | Single Answer | 3111 | 3093 | 17 | 99.42% |
> | Multiple Answer | 3072 | 3057 | 14 | 99.51% |
> | Unanswerable | 4764 | 4749 | 15 | 99.69% |
>
> * * *
> > **Questions 1**: Judger and agreement: Which model/version is used as the judger? What is the agreement with two human annotators on a random subset? What is the variance when replacing the judger with a different model family? (Please include sample size.)
>
> Since this suggestion is similar to Weaknesses 3, we addressed the reviewer's concern about this point in our response to Weaknesses 3.
>
> * * *
> > **Questions 2**: Unanswerable threshold and calibration: What triggers refusal? Did you try confidence calibration or temperature scaling?
>
> Regarding what triggers refusal, the model is instructed first to check whether any candidate satisfies all must-have requirements. If the model concludes that no candidate simultaneously satisfies every must-have, it should output the refusal string.
>
> To address the concern that avoidance might be a sampling artifact, we run an additional study on Doubao-1.6-seed under different decoding temperatures compared to the fully deterministic setting with temperature=0. Accuracy on FTF-Bench under temperatures from 0 to 1 is reported below.
>
> | Temperature | Single-Answer | Multiple-Answer | Unanswerable | Average |
> |-|-|--|--|-|
> | 0.0 | 78.66 | 76.99 | 84.05 | 80.59 |
> | 0.2 | 88.16 | 80.57 | 73.61 | 81.70 |
> | 0.4 | 79.41 | 80.96 | 62.28 | 72.21 |
> | 0.6 | 81.10 | 79.10 | 61.37 | 71.92 |
> | 0.8 | 90.24 | 82.89 | 77.55 | 82.65 |
> | 1.0 | 89.47 | 82.29 | 77.81 | 82.38 |
>
> The overall accuracy varies only moderately across temperatures, indicating that the tendency to refuse is stable.
>
> We further measure the agreement between outputs at temperature 0 and outputs at other temperatures on an instance-by-instance basis.
>
> | Temperature | Single-Answer | Multiple-Answer | Unanswerable | Average |
> |-|-|-|-|-|
> | 0.2 | 79.19 | 75.24 | 80.89 | 77.79 |
> | 0.4 | 74.00 | 73.33 | 67.16 | 70.72 |
> | 0.6 | 73.55 | 73.47 | 66.55 | 70.47 |
> | 0.8 | 79.11 | 75.22 | 78.34 | 77.74 |
> | 1.0 | 78.84 | 75.54 | 79.21 | 78.13 |
>
> This high consistency shows that the refusal behavior and the error patterns we analyze are stable. We have included the results above in Appendix L of the revised version of our paper.
>
> * * *
> > **Questions 3**: Add at least one public benchmark: Please include evaluation on at least one public benchmark.
>
> We thank the reviewer for this suggestion. Our work already evaluates FTF-RL–trained MLLMs on 4 widely used public benchmarks, including LogicVista, MathVision, MathVista and InfoQA. Further, in response to Weakness 1, we have further expanded this evaluation to additional public benchmarks. All these results show that the requirement-aware reasoning transfers beyond FTF-Bench.
>
> * * *
> We hope that our explanations above can clarify your doubts and you can consider our work more favorably.

---

### Official Review · Reviewer_weeR · 2025-11-01

**Soundness:** 3
**Presentation:** 3
**Contribution:** 3
**Rating:** 4
**Confidence:** 4

**Summary:**

This paper proposes a new benchmark and training method for multimodal large language models (MLLMs). It introduces FTF-Bench, a dataset of 3,649 realistic service tasks (e.g., booking, shopping, maps) designed to test whether models can distinguish and prioritize must-have from nice-to-have requirements. Current MLLMs often misinterpret or violate essential constraints, leading to poor reasoning performance. To address this, the authors develop FTF-RL, a multi-objective reinforcement learning framework that rewards correct requirement classification, structured reasoning, and valid outputs. Experiments show that FTF-RL significantly improves task success rates and general reasoning benchmarks (LogicVista, MathVision, etc.). The work highlights requirement-aware reasoning as a key factor in building reliable, generalizable MLLM agents

**Strengths:**

1. The paper introduces FTF-Bench, a large and well-designed benchmark that captures real-world service scenarios with clear distinctions between must-have and nice-to-have requirements. This enables precise evaluation of models’ ability to handle multi-priority reasoning, which previous datasets ignored.
2. The proposed FTF-RL method integrates multi-objective rewards to improve requirement understanding, structured reasoning, and output validity. Experiments demonstrate significant accuracy gains across both in-domain (FTF-Bench) and out-of-domain reasoning benchmarks.
3. The study conducts comprehensive evaluations on proprietary and open-source MLLMs, showing consistent improvements and clear diagnostic insights. Moreover, models fine-tuned with FTF-RL generalize better to unrelated logical and mathematical reasoning tasks, proving its broader effectiveness.

**Weaknesses:**

1. The number of baselines in the experiments is too limited; adding more baseline results would make the experimental section more comprehensive and convincing.
2. The paper’s analysis of why requirement-aware reasoning can enhance general reasoning capabilities is overly vague and unconvincing. In tasks like Math, reasoning intuitively does not involve must-have and nice-to-have requirements, so the contribution of FTF-trained models to general reasoning tasks needs more detailed quantitative and qualitative analysis.
3. There are some inconsistencies in the experimental results section: Section 5.3.2 claims evaluations on LogicVista, MathVision, and MathVista, but in Section 5.3.3, the “four benchmarks” include InfoQA instead of MathVista. This inconsistency raises doubts about whether the performance of FTF-RL on InfoQA (in Table 3) and the Ablation Study on MathVista (in Table 4) aligns with the authors’ analysis.

**Questions:**

1. It would be helpful to show how the model distinguishes between must-have and nice-to-have requirements to better understand its capability boundaries on such tasks, for example by presenting confusion matrices for different purposes.
2. I suggest adding examples and qualitative analyses of the model’s performance on general reasoning benchmarks to explain how FTF-RL’s capabilities transfer to them, further emphasizing the scalability of this work.
3. Providing metric changes during the FTF-RL training process (e.g., answer correctness and classification accuracy) would better highlight the necessity of adopting RL instead of direct SFT in this domain.
4. To more fairly evaluate the effectiveness of FTF-RL, I would like to know whether the 10% evaluation subset contains any bias. Including results of other models on this subset and comparing them with the full dataset outcomes would effectively clarify this point.

---

> ### Author Response · Authors · 2025-11-23
> **Response to Reviewer weeR (Part 1)**
>
> We thank the reviewer for the thoughtful feedback. We address the comments below.
>
> * * *
> > **Weaknesses 1**: The number of baselines in the experiments is too limited; adding more baseline results would make the experimental section more comprehensive and convincing.
>
> We further present a comparative experiment on LLaVA variants, which have undergone various forms of fine-tuning on general-purpose corpora. Results show that they exhibit significant improvements not only in requirement-aware reasoning itself but also in reasoning tasks across other scenarios. We have included the results below in Table 3 and 4 of the revised version of our paper.
>
> Impact of reinforcement learning:
> | Models | Single-Answer | Multiple-Answer | Unanswerable | Average |
> |-|-|-|-|-|
> | LLaVA-OneVision-7B | 45.82 | 33.69 | 42.87 | 44.09 |
> | $\quad$+FTF-RL | **$52.51_{\uparrow 6.69}$** | **$40.15_{\uparrow 6.46}$** | **$48.79_{\uparrow 5.92}$** | **$50.48_{\uparrow 6.39}$** |
> | LLaVA-1.5-13B | 49.23 | 37.51 | 43.19 | 46.64 |
> | $\quad$+FTF-RL | **$57.81_{\uparrow 8.58}$** | **$43.74_{\uparrow 6.23}$** | **$49.05_{\uparrow 5.86}$** | **$52.87_{\uparrow 7.23}$** |
>
> Effect of FTF-RL across general reasoning benchmarks:
> | Models | LogicVista | MathVision | MathVista | InfoQA |
> |-|-|-|-|-|
> | LLaVA-1.5-13B | 29.23 | 11.12 | 27.64 | 41.57 |
> | $\quad$+FTF-RL | **$35.61_{\uparrow 6.38}$** | **$13.74_{\uparrow 2.62}$** | **$29.05_{\uparrow 1.41}$** | **$41.92_{\uparrow 0.35}$** |
>
> In addition, we have also run FTF-Bench on a wider set of baseline models, including the proprietary Claude Sonnet 4.5 and three LLaVA variants. We have included the results below in Table 2 of the revised version of our paper. Across all settings, supplying models with explicit requirements consistently yields higher accuracy than using the original vague user requirements, which further confirms the central claim of our work that requirement-aware reasoning is a missing ingredient.
>
> Interestingly, the gap between vague and explicit requirements is especially pronounced for almost all open-source LLMs, indicating that they generally lack the capability to disentangle vague user requirements into well organized must-have and nice-to-have requirements.
>
> Claude-Sonnet-4.5:
>
> | Scenarios | Single-Answer | Multiple-Answer | Unanswerable | Average |
> |-|-|-|-|-|
> | Upper | **77.52** | **73.89** | 80.98 | **78.01** |
> | Direct | $75.37_{\downarrow 2.15}$ | $71.94_{\downarrow 1.95}$ | **$82.49_{\uparrow 1.51}$** | $77.51_{\downarrow 0.50}$ |
>
> LLaVA-OneVision-7B:
>
> | Scenarios | Single-Answer | Multiple-Answer | Unanswerable | Average |
> |-|-|-|-|-|
> | Upper | **45.28** | **43.17** | **42.36** | **43.38** |
> | Direct | $19.15_{\downarrow 26.13}$ | $18.22_{\downarrow 24.95}$ | $21.78_{\downarrow 20.58}$ | $19.73_{\downarrow 23.65}$ |
>
> LLaVA-1.5-13B:
>
> | Scenarios | Single-Answer | Multiple-Answer | Unanswerable | Average |
> |-|-|-|-|-|
> | Upper | **48.70** | **46.94** | **42.77** | **45.72** |
> | Direct | $26.98_{\downarrow 21.72}$ | $16.39_{\downarrow 30.55}$ | $34.00_{\downarrow 8.77}$ | $25.28_{\downarrow 20.44}$ |
>
> LLaVA-NEXT-13B:
>
> | Scenarios | Single-Answer | Multiple-Answer | Unanswerable | Average |
> |-|-|-|-|-|
> | Upper | **48.65** | **46.89** | **46.72** | **47.25** |
> | Direct | $26.92_{\downarrow 21.73}$ | $18.34_{\downarrow 28.55}$ | $33.95_{\downarrow 12.77}$ | $26.13_{\downarrow 21.12}$ |
>
> LLaVA-NEXT-34B:
>
> | Scenarios | Single-Answer | Multiple-Answer | Unanswerable | Average |
> |-|-|-|-|-|
> | Upper | **50.31** | **50.20** | **51.40** | **50.76** |
> | Direct | $27.27_{\downarrow 23.04}$ | $20.21_{\downarrow 29.99}$ | $25.83_{\downarrow 25.57}$ | $24.11_{\downarrow 26.65}$ |

---

> ### Author Response · Authors · 2025-11-23
> **Response to Reviewer weeR (Part 2)**
>
> * * *
> > **Weaknesses 2**: The paper’s analysis of why requirement-aware reasoning can enhance general reasoning capabilities is overly vague and unconvincing. In tasks like Math, reasoning intuitively does not involve must-have and nice-to-have requirements, so the contribution of FTF-trained models to general reasoning tasks needs more detailed quantitative and qualitative analysis.
>
> Learning to separate must-have from nice-to-have requirements is essentially learning to distinguish necessary conditions from preference-like or secondary conditions, and to enforce the former before optimizing the latter. This maps naturally to a broad range of reasoning tasks that are not framed as user requests. FTF-RL explicitly trains MLLMs to first construct a structured requirement parse instead of mixing these steps in a pattern-matching way. This inductive bias toward explicit requirement extraction is task-agnostic.
>
> Beyond intuition, our results provide quantitative evidence that the improvements are not merely accidental occurrences on FTF-Bench. First, FTF-RL shows consistent gains on various general benchmarks after FTF-RL (such as the 4 benchmarks LogicVista, MathVision, MathVista, InfoQA in the original paper), while all training signals come only from requirement-aware tasks.
>
> We further evaluate models fine tuned with FTF-RL on two additional public benchmarks. AndroidControl represents complex software control in an operating system environment. ScienceQA captures multi-step scientific reasoning.
>
> Across both benchmarks we observe consistent improvements after FTF-RL, mirroring the gains we reported previously on LogicVista, MathVision, and MathVista. This shows that learning to prioritize requirements on FTF-Bench transfers to more complex agent behaviors.
>
> | Model | $\text{ScienceQA}$ |
> |-|-|
> | Qwen2.5 VL 7B Instruct | 40.06 |
> | $\quad$+FTF-RL | $40.28_{\uparrow 0.22}$ |
>
> | Model | Metric | CLICK | TYPE | SCROLL | OPENAPP | WAIT | COMPLETE | PRESS |
> |-|-|-|-|-|-|-|-|-|
> | Qwen2.5-VL-7B-Instruct | TMR | 0.9538 | 0.8880 | 0.8571 | 0.0000 | 0.0459 | 0.9339 | 0.2857 |
> | Qwen2.5-VL-7B-Instruct | AMR | 0.3317 | 0.6800 | 0.0603 | 0.0000 | 0.0459 | 0.9339 | 0.2857 |
> | $\quad$+FTF-RL | TMR | $0.9608_{\uparrow 0.0071}$ | $0.9216_{\uparrow 0.0336}$ | $0.8026_{\downarrow 0.0545}$ | $0.000_{\uparrow 0.0000}$ | $0.0529_{\uparrow 0.0071}$ | $0.9533_{\uparrow 0.0194}$ | $0.5364_{\uparrow 0.2507}$ |
> | $\quad$+FTF-RL | AMR | $0.3240_{\downarrow 0.0077}$ | $0.7184_{\uparrow 0.0384}$ | $0.0446_{\downarrow 0.0157}$ | $0.000_{\uparrow 0.0000}$ | $0.0529_{\uparrow 0.0071}$ | $0.9533_{\uparrow 0.0194}$ | $0.5364_{\uparrow 0.2507}$ |
>
> Finally, in response to Question 2, we add two case studies based on Qwen2.5-VL-7B-Instruct with FTF-RL and place the full examples in a new Appendix I. These examples complement the quantitative improvements in Table 4 of our paper and illustrate that FTF-RL does not merely overfit to the style of FTF-Bench. Instead, it encourages the trained model to first parse the essential requirement of a problem and then perform reasoning.
>
> * * *
> > **Weaknesses 3**: There are some inconsistencies in the experimental results section: Section 5.3.2 claims evaluations on LogicVista, MathVision, and MathVista, but in Section 5.3.3, the “four benchmarks” include InfoQA instead of MathVista. This inconsistency raises doubts about whether the performance of FTF-RL on InfoQA (in Table 3) and the Ablation Study on MathVista (in Table 4) aligns with the authors’ analysis.
>
> We thank the reviewer for the thoughtful feedback. We have included the extended experimental results in Section 5.3.2 of the revised version. The updated section now reports FTF-RL performance on InfoQA so that it is fully aligned with analysis in Section 5.3.3. We have also added LLaVA-1.5-13B to further validate our claim.
>
> It can be seen that models trained with FTF-RL achieve improvements on the majority of reasoning tasks, even if the training is performed only on FTF-Bench. This suggests that requirement-aware reasoning not only strengthens the understanding of complex user intent but also stimulates the general reasoning ability of MLLMs, leading to clear gains across diverse challenging tasks.
>
> | Model | LogicVista | MathVision | MathVista | InfoQA |
> |-|-|-|-|-|
> | Qwen2.5-VL-7B-Instruct | $43.40$ | $24.67$ | $60.80$ | 65.35 |
> | $\quad$+FTF-RL | **$47.43_{\uparrow 4.03}$** | **$25.65_{\uparrow 0.98}$** | **$61.30_{\uparrow 0.05}$** | **$67.81_{\uparrow 2.46}$** |
> | Qwen2.5-VL-3B-Instruct | $36.91$ | $23.03$   | **$40.70$** | 37.62 |
> | $\quad$+FTF-RL | **$40.49_{\uparrow 3.58}$** | **$24.34_{\uparrow 1.31}$** | $39.50_{\downarrow 1.20}$ | **$39.93{\uparrow 2.31}$** |
> | LLaVA-1.5-13B | $29.23$ | $11.12$ | $27.64$ | 41.57 |
> | $\quad$+FTF-RL | **$35.61_{\uparrow 6.38}$** | **$13.74_{\uparrow 2.62}$** | **$29.05_{\uparrow 1.41}$** | **$41.92_{\uparrow 0.35}$** |

---

> ### Author Response · Authors · 2025-11-23
> **Response to Reviewer weeR (Part 3)**
>
> * * *
> > **Questions 1**: It would be helpful to show how the model distinguishes between must-have and nice-to-have requirements to better understand its capability boundaries on such tasks, for example by presenting confusion matrices for different purposes.
>
> For each requirement in FTF Bench, we directly compare the annotated label with the label predicted by the model under the Direct setting and aggregate the counts into a confusion matrix. We have included the results below in Appendix G of the revised version of our paper.
>
> Doubao-1.6-seed:
>
> | Ground truth / Prediction | Predicted must-have | Predicted nice-to-have | Total |
> |-|-|-|-|
> | Must-have | 5221 (64.74%) | 2844 (35.26%) | 8065 |
> | Nice-to-have  | 2068 (16.79%) | 10246 (83.21%) | 12314 |
> | Total | 7289 | 13090 | 20379 |
>
> GPT-5 requirement classification (accuracy 85.14)
>
> | Ground truth / Prediction | Predicted must-have | Predicted nice-to-have | Total |
> |-|-|-|-|
> | Must-have | 6425 (79.67%) | 1640 (20.33%) | 8065 |
> | Nice-to-have | 1418 (11.52%) | 10896 (88.48%) | 12314 |
> | Total | 7843 | 12486 | 20379 |
>
> Qwen2.5-VL-7B-Instruct requirement classification (accuracy 59.65)
>
> | Ground truth / Prediction | Predicted must-have | Predicted nice-to-have | Total |
> |-|-|-|-|
> | Must-have | 4073 (50.52%) | 3992 (49.48%) | 8065 |
> | Nice-to-have | 4229 (34.34%) | 8085 (65.66%) | 12314 |
> | Total | 7843 | 12486 | 20379 |
>
> It can be seen that GPT-5 behaves in a relatively balanced way for both must-have and nice-to-have requirements. Doubao-1.6-seed shows strong recognition of nice-to-have requirements, but it also tends to classify more must-have requirements as nice-to-have. Qwen2.5-VL-7B-Instruct has a generally weaker capability to correctly separate the two types of requirements. We have included these confusion matrices and the interesting findings in Appendix G of the revised version.
>
> * * *
> > **Questions 2**: I suggest adding examples and qualitative analyses of the model’s performance on general reasoning benchmarks to explain how FTF-RL’s capabilities transfer to them, further emphasizing the scalability of this work.
>
> In the revised version, we add two case studies based on Qwen2.5-VL-7B-Instruct with FTF-RL and place the full examples in Appendix I.
>
> The first case is a MathVision geometry problem about a circular carpet on a tiled floor. The model must decide which grey tile pattern cannot come from any circle. After FTF-RL, Qwen2.5-VL-7B-Instruct does not jump directly to an option. It first summarizes the task requirement that grey tiles must be exactly those intersected by a single convex circle, so they should form one connected region with a smooth boundary. It then explicitly plans to check each candidate against this requirement and finally concludes that the option with a disconnected grey region is impossible. This shows that the model uses a requirement-driven plan rather than local pattern matching.
>
> The second case is a LogicVista problem that involves inferring the meaning of two symbolic operations applied to shapes. The model must fill in a missing output shape and a missing operation symbol. After FTF-RL, Qwen2.5-VL-7B-Instruct begins by restating the subgoals, then applies these inferred rules to the two queries. It then follows this plan step by step and correctly selects the option. The model learns to organize the task into requirement extraction and execution.
>
> These examples complement the quantitative improvements in Table 4 of our paper and illustrate that FTF-RL does not merely overfit to the style of FTF-Bench. Instead, it encourages the trained model to first parse the essential requirement of a problem and then perform reasoning. In the revised version, we describe these two cases and provide full versions, including original questions, images, and model outputs in Appendix I.
>
> * * *
> > **Questions 3**: Providing metric changes during the FTF-RL training process (e.g., answer correctness and classification accuracy) would better highlight the necessity of adopting RL instead of direct SFT in this domain.
>
> We thank the reviewer for the suggestion. We further record the evolution of different rewards over the whole FTF-RL training process on Qwen2.5-VL-7B-Instruct below:
>
> | Metric / Step | 1 | 10 | 20 | 30 | 40 | 50 | 60 | 70 | 75 |
> |-|-|-|-|-|-|-|-|-|-|
> | reward/format | 0.781 | 0.982 | 0.989 | 0.992 | 0.993 | 0.994 | 0.996 | 0.996 | 0.997 |
> | reward/requirement_accuracy | 0.601 | 0.802 | 0.887 | 0.903 | 0.921 | 0.932 | 0.943 | 0.949 | 0.952 |
> | reward/accuracy | 0.182 | 0.422 | 0.521 | 0.613 | 0.676 | 0.715 | 0.748 | 0.762 | 0.775 |
> | reward/overall | 0.353 | 0.587 | 0.672 | 0.741 | 0.789 | 0.813 | 0.835 | 0.857 | 0.858 |
>
> It shows consistent improvements in answer correctness as training proceeds. We have also included the full reward-versus-step curves in Appendix H of the revised version of our paper.

---

> ### Author Response · Authors · 2025-11-23
> **Response to Reviewer weeR (Part 4)**
>
> * * *
> > **Questions 4**: To more fairly evaluate the effectiveness of FTF-RL, I would like to know whether the 10% evaluation subset contains any bias. Including results of other models on this subset and comparing them with the full dataset outcomes would effectively clarify this point.
>
> In the RL experiments, we randomly sample 90% of FTF-Bench for training and hold out the remaining 10% for evaluation, and the same subset is used for all models. To check whether this subset introduces bias, we have now evaluated all baseline models on both the full benchmark and this 10% subset and compared the results. We have also extended the results in Appendix K of the revised version.
>
> | Model | Split | Single Upper | Single Direct | Multiple Upper | Multiple Direct | Unanswerable Upper | Unanswerable Direct | Average Upper | Average Direct |
> |-|-|-|-|-|-|-|-|-|-|
> | Gemini-2.5-pro | Full | 88.89 | 86.91 | 84.20 | 82.26 | 81.72 | 78.55 | 84.26 | 81.75 |
> | Gemini-2.5-pro | Subset  | 79.61 | 81.55 | 72.28 | 69.31 | 63.29 | 58.23 | 69.16 | 66.57 |
> | GPT-o3 | Full | 82.67 | 77.78 | 80.41 | 80.03 | 83.31 | 83.09 | 82.33 | 79.68 |
> | GPT-o3 | Subset | 75.73 | 72.15 | 69.31 | 75.25 | 81.01 | 73.79 | 75.22 | 72.33 |
> | Doubao-1.6-seed | Full | 80.09 | 78.66 | 81.70 | 76.99 | 83.80 | 84.05 | 82.24 | 80.59 |
> | Doubao-1.6-seed | Subset | 72.82 | 66.99 | 63.37 | 58.42 | 74.68 | 68.99 | 69.74 | 63.98 |
> | LLaMA-4 | Full | 58.36 | 55.48 | 55.93 | 54.45 | 44.43 | 36.94 | 55.03 | 51.99 |
> | LLaMA-4 | Subset | 54.37 | 49.51 | 41.58 | 25.32 | 30.38 | 40.59 | 37.75 | 33.72  |
> | Qwen2.5-VL-7B-Instruct | Full | 61.16 | 22.99 | 58.25 | 20.10 | 47.59 | 18.64 | 57.69 | 21.05 |
> | Qwen2.5-VL-7B-Instruct | Subset | 57.28 | 20.39 | 39.60 | 11.88 | 37.34 | 9.49  | 41.21 | 9.51 |
> | Qwen2.5-VL-32B-Instruct | Full | 70.66 | 69.25 | 70.55 | 70.17 | 53.08 | 50.58 | 67.72 | 66.57  |
> | Qwen2.5-VL-32B-Instruct | Subset | 66.99 | 67.96 | 42.41 | 34.18 | 54.46 | 58.42 | 50.72 | 48.41 |
>
> Across all models, we expand the results into six scenario specific accuracies for each model. **Treating each (model, scenario) pair as one data point, the Pearson correlation between accuracies on the full benchmark and on the 10% subset is 0.895, and the Kendall correlation is 0.708.** Although scores on the subset are slightly lower on average, the strong linear and rank correlations show that the relative ordering of models and the gap between them are highly consistent across the two splits.
>
> * * *
> We hope that our explanations above can clarify your doubts and you can consider our work more favorably.

---

### Meta-Review · Area_Chair_zi9y · 2026-01-08

**Summary:**

This work proposes a benchmark and approach to evaluate and train multimodal large language models (MLLM) for complex, structured user requirements. It receives initial review scores of 1 borderline accept, 2 borderline rejects, and 1 reject. Reviewers raised concerns on the insufficient experiments, unconvincing analysis, inconsistent results, narrowness of task definition and testing scenarios, and limited innovation.

**Reviewer Concerns:**

Reviewer weeR's most concerns are addressed by experiments, but the AC feels that the explanation to W2 "why requirement-aware reasoning can enhance general reasoning capabilities is overly vague and unconvincing" may not be sufficient enough.

Reviewer Dr9c's concern on task narrowness is explained during rebuttal, but may not be convincing enough to the reviewer. Other concerns of Reviewer Dr9c are addressed by experimental results.

Reviewer Gh4d's concerns on comparative experiments, incomplete ablation, and limited innovation are answered in the rebuttal. Some additional experiments are used to address these questions, but may not be enough to address the reviewer's concerns, especially the critical question of limited innovation.

Reviewer SuaU indicated that the rebuttal addressed his/her questions and that he/she will maintain the original score of borderline accept.

**Reviewer Scores:**

Reviewer weeR and Dr9c will still keep a borderline score, either maintaining borderline reject or increase to borderline accept.

Reviewer Gh4d will likely keep the reject score.

Reviewer SuaU mentioned that he/she will keep the original borderline accept score.

---

### Decision · Program_Chairs · 2026-01-26

Reject